## PROCEEDINGS A

rspa.royalsocietypublishing.org

# Research

applied mathematics, differential equations, mathematical modelling

model predictive control, nonlinear dynamics, sparse identification of nonlinear dynamics, system identification, control theory, machine learning

**Author for correspondence:**
E. Kaiser
e-mail: eurika@uw.edu

# Sparse identification of nonlinear dynamics for model predictive control in the low-data limit

E. Kaiser[1], J. N. Kutz[2] and S. L. Brunton[1]

[1]Department of Mechanical Engineering, and [2]Department of Applied Mathematics, University of Washington, Seattle, WA, 98195

EK, 0000-0001-6049-0812

Data-driven discovery of dynamics via machine learning is pushing the frontiers of modelling and control efforts, providing a tremendous opportunity to extend the reach of model predictive control (MPC). However, many leading methods in machine learning, such as neural networks (NN), require large volumes of training data, may not be interpretable, do not easily include known constraints and symmetries, and may not generalize beyond the attractor where models are trained. These factors limit their use for the online identification of a model in the low-data limit, for example following an abrupt change to the system dynamics. In this work, we extend the recent sparse identification of nonlinear dynamics (SINDY) modelling procedure to include the effects of actuation and demonstrate the ability of these models to enhance the performance of MPC, based on limited, noisy data. SINDY models are parsimonious, identifying the fewest terms in the model needed to explain the data, making them interpretable and generalizable. We show that the resulting SINDY-MPC framework has higher performance, requires significantly less data, and is more computationally efficient and robust to noise than NN models, making it viable for online training and execution in response to rapid system changes. SINDY-MPC also shows improved performance over linear data-driven models, although linear models may provide a stopgap until enough data is available for SINDY. SINDY-MPC is demonstrated on a variety of dynamical systems with different challenges,

including the chaotic Lorenz system, a simple model for flight control of an F8 aircraft, and an HIV model incorporating drug treatment.

## 1. Introduction

Data-fuelled modelling and control of complex systems is currently undergoing a revolution, driven by the confluence of big data, advanced algorithms in machine learning and modern computational hardware. Model-based control strategies, such as model predictive control (MPC), are ubiquitous, relying on accurate and efficient models that capture the relevant dynamics for a given objective. Increasingly, first principles models are giving way to data-driven approaches, for example in turbulence, epidemiology, neuroscience and finance [1]. Although these methods offer tremendous promise, there has been slow progress in distilling physical models of dynamic processes from data. Despite their undeniable success, many modern techniques in machine learning (e.g. neural networks (NN)) rely on access to massive datasets, have limited ability to generalize beyond the attractor where data is collected, and do not readily incorporate known physical constraints. The current challenges associated with data-driven discovery limit its use for real-time control of strongly nonlinear, high-dimensional, multi-scale systems, and prevent online recovery in response to abrupt changes in the dynamics. Fortunately, a new paradigm of sparse and parsimonious modelling is enabling interpretable models in the low-data limit. In this work, we extend the recent sparse identification of nonlinear dynamics (SINDy) framework [2] to identify models with actuation, and combine it with MPC for effective and interpretable data-driven, model-based control. We apply the proposed SINDY-MPC architecture to control several nonlinear systems and demonstrate improved control performance in the low-data limit, compared with other leading data-driven methods, including linear response models and NNs.

Model-based control techniques, such as MPC [3,4] and optimal control [5,6], are cornerstones of advanced process control, and are well-positioned to take advantage of the data-driven revolution. MPC is particularly ubiquitous in industrial applications, as it enables the control of strongly nonlinear systems with constraints, which are difficult to handle using traditional linear control approaches [7–11]. MPC benefits from simple and intuitive tuning and the ability to control a range of simple and complex phenomena, including systems with time delays, non-minimum phase dynamics, and instability. In addition, it is straightforward to incorporate known constraints and multiple operating conditions, it exhibits an intrinsic compensation for dead time, and it provides the flexibility to formulate and tailor a control objective. The major drawback of model-based control, such as MPC, lies in the development of a suitable model via existing system identification or model reduction techniques [12], which may require expensive and time-consuming data collection and computations.

Nearly all industrial applications of MPC rely on empirical models, and increasing plant complexity and tighter performance specifications require models with higher accuracy. There are many techniques to obtain data-driven models, including state-space models from the eigensystem realization algorithm (ERA) [13] and other subspace identification methods, Volterra series [14–16], autoregressive models [17] (e.g. ARX, ARMA, NARX and NARMAX [18] models), and NN models [19–22], to name only a few. These procedures all tend to yield black-box models, with limited interpretability, physical insights and ability to generalize. More recently, linear representations of nonlinear systems using extended dynamic mode decomposition [23] have been successfully paired with MPC [24,25]. Nonlinear models based on machine learning, such as NNs, are increasingly used due to advances in computing power, and recently deep reinforcement learning has been combined with MPC [26,27], yielding impressive results in the large-data limit. However, large volumes of data are often a luxury, and many systems must be identified and controlled with limited data, for example in response to abrupt changes. Current efforts are focused on *rapid* learning based on minimal data.

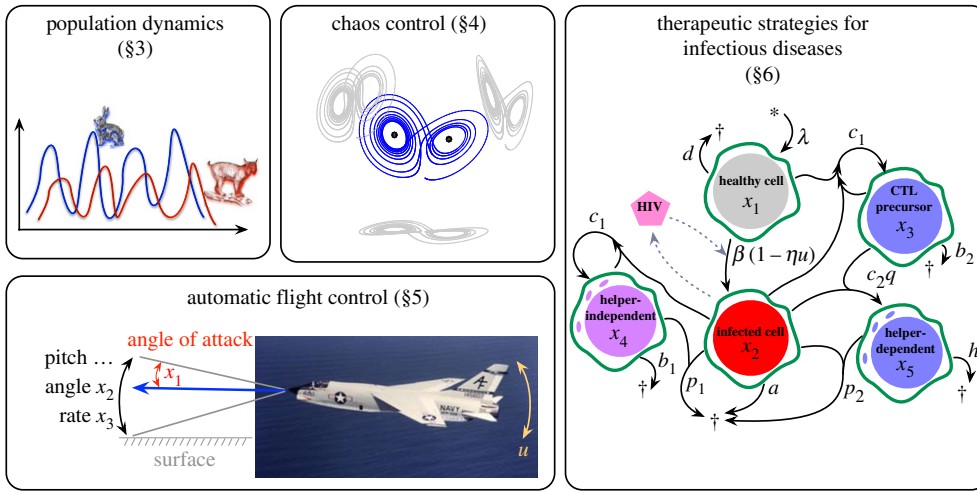

**Figure 1.** Applications of SINDY-MPC investigated in this work. (Online version in colour.)

When abrupt changes occur in the system, an effective controller must rapidly characterize and compensate for the new dynamics, leaving little time for discovery based on limited data. A second challenge is the ability of models to generalize beyond the training data, which is related to the ability to incorporate new information and quickly modify the model. Machine learning algorithms often suffer from overfitting and a lack of interpretability, although the application of these algorithms to physical systems offers a unique opportunity to incorporate known symmetries and constraints. These challenges point to the need for *parsimonious* and interpretable models [2,28,29] that may be characterized from limited data and in response to abrupt changes [30]. Whereas traditional methods require unrealistic amounts of training data, the recently proposed SINDY framework [2] relies on sparsity-promoting optimization to identify parsimonious models from limited data, resulting in interpretable models that avoid overfitting. It has also been shown recently [31] that it is possible to enforce known physics (e.g. constraints, conservation laws and symmetries) in the SINDY algorithm, improving stability and performance of models.

In this work, we combine SINDY with MPC for enhanced data-driven control of nonlinear systems in the low-data limit. First, we extend the SINDY architecture to identify interpretable models that include nonlinear dynamics and the effect of actuation. Next, we show the enhanced performance of SINDY-MPC compared with linear data-driven models and with NN models. The linear models are identified using dynamic mode decomposition with control (DMDc) [1,32], which is closely related to SINDY and traditional state-space modelling techniques such as ERA. SINDY-MPC is shown to have better prediction accuracy and control performance than NN models, especially for small and moderate amounts of noisy data. In addition, SINDY models are less expensive to train and execute than NN models, enabling real-time applications. SINDY-MPC also outperforms linear models for moderate amounts of data, although DMDc provides a working model in the extremely low-data limit for simple problems. Thus, in response to abrupt changes, a linear DMDc model may be used until a more accurate SINDY model is trained. We demonstrate the SINDY-MPC architecture on several systems of increasing complexity as illustrated in figure 1.

## 2. SINDY-MPC framework

The SINDY-MPC architecture combines the systematic data-driven discovery of dynamics with advanced model-based control to facilitate rapid model learning and control of strongly nonlinear

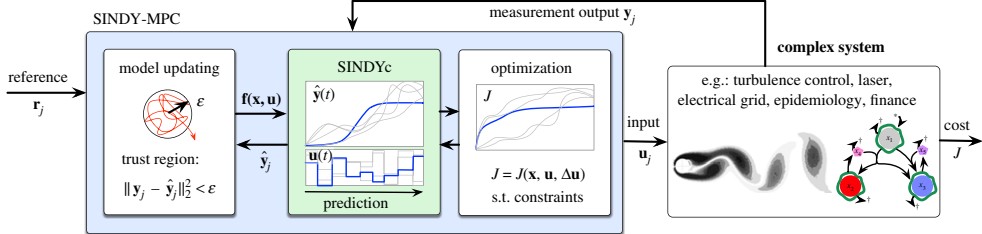

**Figure 2.** Schematic of the proposed SINDY-MPC framework, using sparse nonlinear models for predictive control. (Online version in colour.)

systems. The overarching SINDY-MPC framework is illustrated in figure 2. In the following sections, we will describe the sparse identification of nonlinear dynamics with control and MPC algorithms. We consider the nonlinear dynamical system

$$\frac{\mathrm{d}}{\mathrm{d}t}\mathbf{x} = \mathbf{f}(\mathbf{x}, \mathbf{u}), \quad \mathbf{x}(0) = \mathbf{x}_0, \tag{2.1}$$

with state $\mathbf{x} \in \mathbb{R}^n$, control input $\mathbf{u} \in \mathbb{R}^q$ and smooth dynamics $\mathbf{f}(\mathbf{x}, \mathbf{u}) : \mathbb{R}^n \times \mathbb{R}^q \to \mathbb{R}^n$.

## (a) Sparse identification of nonlinear dynamics with control

Advanced machine learning algorithms provide new opportunities for nonlinear system identification. In particular, sparsity-promoting methods are playing an increasingly important role by recognizing the importance of parsimony in models [2,33,34], i.e. the trade-off between model complexity and data fit. Recent work based on compressed sensing has been used to handle noise and outliers [35] for linear system identification and large libraries of candidate functions [36]. Sparse regularization, which has been demonstrated for parameter and structure identification [2,34,37,38], is a particularly promising direction as this can promote robustness and generalizability in models. We refer the reader to an extensive review on nonlinear system identification methods [39] and a recent review in the context of machine learning [40].

Here, we generalize the sparse identification of nonlinear dynamics (SINDY) method [2] to include inputs and control as illustrated in figure 3. SINDY identifies nonlinear dynamical systems from measurement data, relying on the fact that many systems have relatively few terms in the governing equations. Thus, sparsity-promoting techniques may be used to find models that automatically balance sparsity in the number of model terms with accuracy, resulting in parsimonious models. In particular, a library of candidate nonlinear terms $\boldsymbol{\Theta}(\mathbf{x})$ is constructed, and sparse regression is used to identify the few active terms in $\boldsymbol{\Theta}$ to approximate the function $\mathbf{f}$.

SINDY with control (SINDYc) is based on the same assumption, that equation (2.1) only has a few active terms in the dynamics. SINDY is readily generalized to include actuation, as this merely requires a larger library $\boldsymbol{\Theta}(\mathbf{x}, \mathbf{u})$ of candidate functions that include $\mathbf{u}$; these functions can include nonlinear cross terms in $\mathbf{x}$ and $\mathbf{u}$. Thus, we measure $m$ snapshots of the state $\mathbf{x}$ and the input signal $\mathbf{u}$ in time and arrange these into two matrices:

$$\mathbf{X} = [\mathbf{x}_1 \quad \mathbf{x}_2 \quad \cdots \quad \mathbf{x}_m] \quad \text{and} \quad \mathbf{U} = [\mathbf{u}_1 \quad \mathbf{u}_2 \quad \cdots \quad \mathbf{u}_m]. \tag{2.2}$$

The library of candidate nonlinear functions $\boldsymbol{\Theta}$ may now be evaluated using the data $\mathbf{X}$ and $\mathbf{U}$:

$$\boldsymbol{\Theta}(\mathbf{X}, \mathbf{U}) = [\mathbf{1}^{\mathrm{T}} \ \mathbf{X}^{\mathrm{T}} \ \mathbf{U}^{\mathrm{T}} \ (\mathbf{X} \otimes \mathbf{X})^{\mathrm{T}} \ (\mathbf{X} \otimes \mathbf{U})^{\mathrm{T}} \ \cdots \ \sin(\mathbf{X})^{\mathrm{T}} \sin(\mathbf{U})^{\mathrm{T}} \ \sin(\mathbf{X} \otimes \mathbf{U})^{\mathrm{T}} \ \cdots], \tag{2.3}$$

where $\mathbf{x} \otimes \mathbf{y}$ defines the vector of all product combinations of the components in $\mathbf{x}$ and $\mathbf{u}$. Although this definition includes repeated rows in $\boldsymbol{\Theta}$, in practice, the implementation is restricted to unique combinations. A suitable library of candidate terms is crucial in the SINDYc algorithm. One strategy is to start with a basic choice, such as polynomials, and increase the complexity of the

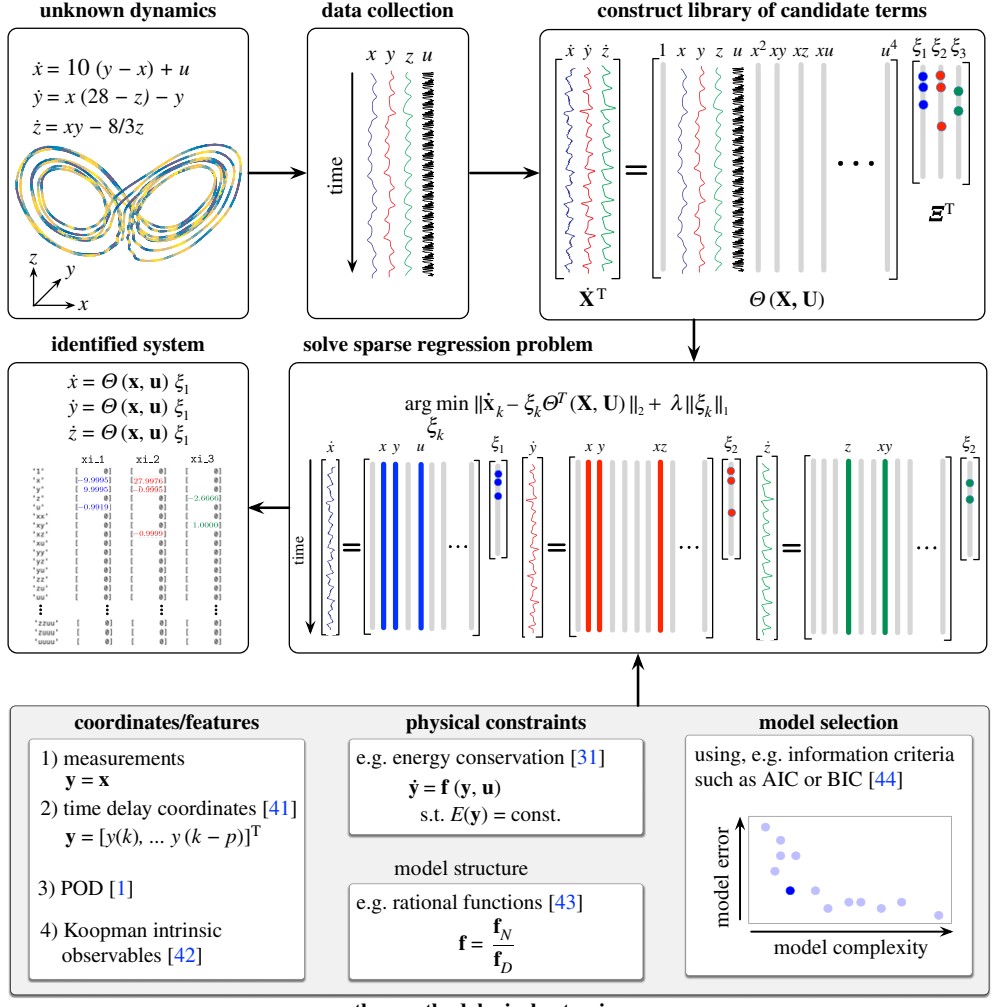

**Figure 3.** Schematic of the SINDYc algorithm and extensions. Active terms in a library of candidate nonlinearities are selected via sparse regression. Illustration of the modular nature of the SINDY with control framework (bottom row) and its ability to handle high-dimensional systems, limited measurements, known physical constraints and model selection. (Online version in colour.)

library by including other terms (trigonometric functions, etc.). It is also possible to incorporate partial knowledge of the physics (e.g. fluids versus quantum mechanics) to decide on a library.

The system in equation (2.1) can thus be written as:

$$\dot{\mathbf{X}} = \boldsymbol{\Xi}\,\boldsymbol{\Theta}^{\mathrm{T}}(\mathbf{X}, \mathbf{U}). \tag{2.4}$$

The time derivatives $\dot{\mathbf{X}} = [\dot{\mathbf{x}}_1\ \dot{\mathbf{x}}_2\ \cdots\ \dot{\mathbf{x}}_m]$, if not measured directly, are computed by numerical differentiation or approximated using the total variation regularized derivative [41,42] if the data is noise-corrupted. The coefficients $\boldsymbol{\Xi}$ are *sparse* for many dynamical systems. Therefore, we employ sparse regression to identify a sparse $\boldsymbol{\Xi}$ corresponding to the fewest nonlinearities in our library that give good model performance:

$$\boldsymbol{\xi}_k = \underset{\hat{\boldsymbol{\xi}}_k}{\mathrm{argmin}}\ \frac{1}{2}\|\dot{\mathbf{X}}_k - \hat{\boldsymbol{\xi}}_k\boldsymbol{\theta}^{\mathrm{T}}(\mathbf{X}, \mathbf{U})\|_2^2 + \lambda\|\hat{\boldsymbol{\xi}}_k\|_1, \tag{2.5}$$

where $\dot{\mathbf{X}}_k$ represents the $k$th row of $\dot{\mathbf{X}}$ and $\boldsymbol{\xi}_k$ is the $k$th row of $\boldsymbol{\Xi}$.

The $\| \cdot \|_1$ term promotes sparsity in the coefficient vector $\boldsymbol{\xi}_k$. This optimization may be solved using the LASSO [43] or the sequentially thresholded least squares procedure [2] (see algorithm 1.). General conditions for the uniqueness of the $l_1$ relaxed solution have been provided in [44]. In practice, these conditions may not be readily met, and false discoveries may occur, although they may be avoided under certain conditions [45]. Specific conditions under which the sequentially thresholded least-squares algorithm in SINDy converges are provided in [46]. More recently, convergence and recovery has been explored in a generalized framework for sparse relaxed regularized regression [47], for which SINDy constitutes a special case. Conditions under which a model structure can be recovered from input–output data have also been examined in the context of identifiability [48,49].

---

**Algorithm 1. Sequentially thresholded least squares to learn the active library components.**

---

**Input:** Time derivative $\dot{\mathbf{X}}$, library of candidate functions $\boldsymbol{\theta}^{\mathrm{T}}(\mathbf{X}, \mathbf{U})$, thresholding parameter $\varepsilon$
**Output:** Matrix of sparse coefficient vectors $\boldsymbol{\Xi}$

1: **function** STLS_REGRESSION$(\dot{\mathbf{X}}, \boldsymbol{\theta}^{\mathrm{T}}(\mathbf{X}, \mathbf{U}), \varepsilon, N)$
2: $\hat{\boldsymbol{\Xi}}^0 \leftarrow (\boldsymbol{\theta}^{\mathrm{T}})^{\dagger} \dot{\mathbf{X}}$ ▷ Initial least squares guess.
3: **while** not converged **do**
4: $k \leftarrow k + 1$
5: $\mathbf{I}_{\mathrm{small}} \leftarrow (\mathrm{abs}(\hat{\boldsymbol{\Xi}}) < \varepsilon)$ ▷ Find small entries ...
6: $\hat{\boldsymbol{\Xi}}^k(\mathbf{I}_{\mathrm{small}}) \leftarrow 0$ ▷ ... and threshold.
7: **for** all variables **do**
8: $\mathbf{I}_{\mathrm{big}} \leftarrow \sim \mathbf{I}_{\mathrm{small}}(:, ii)$ ▷ Find big entries ...
9: $\hat{\boldsymbol{\Xi}}^k(\mathbf{I}_{\mathrm{big}}, ii) \leftarrow (\boldsymbol{\theta}^{\mathrm{T}}(:, \mathbf{I}_{\mathrm{big}}))^{\dagger} \dot{\mathbf{X}}(:, ii)$ ▷ ... and regress onto those terms.
10: **end for**
11: **end while**
12: **end function**

---

The parameter $\lambda$ (or equivalently $\varepsilon$ in algorithm 1.) is selected to identify the Pareto optimal model that best balances model complexity with accuracy. A coarse sweep of $\lambda$ is performed to identify the rough order of magnitude where terms are eliminated and where error begins to increase. Then this parameter sweep may be refined, and the models on the Pareto front are evaluated using information criteria [50]. It is interesting to note, that a similar idea, identifying active components in $\mathbf{f}$ from a library of candidate functions using sparse regularization, was discarded in favour of a Bayesian formulation, as the non-orthogonality of the columns in the library was seen as problematic [38]. However, as in [2], we will demonstrate here the effectiveness of the approach.

Since the original SINDY paper [2], it has been extended to include constraints and known physics [31], for example, to enforce energy preserving constraints in an incompressible fluid flow. SINDY has also been extended to high-dimensional systems, by identifying dynamics on principal components [2], learning partial differential equations [51,52] and extracting dynamics on delay coordinates [53]. Robust variants of SINDY have been formulated to identify models despite large outliers and noise [54,55].

## (i) Discovering discrete-time dynamics

In the original SINDY algorithm, it was shown that it is possible to identify discrete-time models of the form $\mathbf{x}_{k+1} = \mathbf{F}(\mathbf{x}_k)$. It is also possible to extend SINDY to identify discrete-time models with inputs and control:

$$\mathbf{x}_{k+1} = \mathbf{F}(\mathbf{x}_k, \mathbf{u}_k). \tag{2.6}$$

rspa.royalsocietypublishing.org *Proc. R. Soc. A* **474**: 20180335

Instead of computing derivatives, we collect a matrix $\mathbf{X}'$ with the columns of $\mathbf{X}$ advanced one timestep: $\mathbf{X}' = [\mathbf{x}_2 \ \mathbf{x}_3 \ \cdots \ \mathbf{x}_{m+1}]$. Then, the dynamics may be written as

$$\mathbf{X}' = \boldsymbol{\Xi}\boldsymbol{\Theta}^{\mathrm{T}}(\mathbf{X}, \mathbf{U}), \tag{2.7}$$

and the regression problem becomes

$$\boldsymbol{\xi}_k = \underset{\hat{\boldsymbol{\xi}}_k}{\operatorname{argmin}} \frac{1}{2}\|\mathbf{X}'_k - \hat{\boldsymbol{\xi}}_k\boldsymbol{\Theta}^{\mathrm{T}}(\mathbf{X}, \mathbf{U})\|_2^2 + \lambda\|\hat{\boldsymbol{\xi}}_k\|_1. \tag{2.8}$$

## (ii) Relationship to dynamic mode decomposition

The SINDY regression is related to the dynamic mode decomposition (DMD), which originated in the fluids community to extract spatiotemporal coherent structures from large fluid datasets [1,56–58]. DMD modes are spatially coherent and oscillate at a fixed frequency and/or growth or decay rate. Since fluids data are typically high-dimensional, DMD is built on the proper orthogonal decomposition (POD) [59], effectively recombining POD modes in a linear combination to enforce the temporal coherence. The dynamic mode decomposition has been applied to a wide range of problems including fluid mechanics, epidemiology, neuroscience, robotics, finance and video processing [1]. Many of these applications have the ultimate goal of closed-loop feedback control.

In DMD, a similar regression is performed to identify a linear discrete-time model $\mathbf{A}$ mapping $\mathbf{X}$ to $\mathbf{X}'$:

$$\mathbf{X}' = \mathbf{A}\mathbf{X}. \tag{2.9}$$

Thus, SINDY reduces to DMD if formulated in discrete-time, with linear library elements in $\boldsymbol{\Theta}$, and without a sparsity-promoting $L_1$ penalty term, i.e. $\lambda = 0$.

DMD was recently extended to include actuation inputs by Proctor *et al.* [32], to disambiguate the effect of internal dynamics and control. In dynamic mode decomposition with control (DMDc), a similar regression is formed, but with the actuation input matrix $\mathbf{U}$:

$$\mathbf{X}' = \mathbf{A}\mathbf{X} + \mathbf{B}\mathbf{U}. \tag{2.10}$$

Thus, SINDY with control similarly reduces to DMDc under certain conditions. In this work, we will use DMDc and SINDYc to discover dynamics for MPC. The DMDc algorithm has also been shown to be related to other subspace identification methods, such as the ERA [13], but designed for high-dimensional input–output data.

It is interesting to note that the extended DMD (eDMD) [23] regression is performed on the nonlinear library $\boldsymbol{\Theta}(\mathbf{X}') = \mathbf{A}\boldsymbol{\Theta}(\mathbf{X})$, and an $l_1$ penalty may also be added. eDMD may also be modified to incorporate actuation inputs, and these models have recently been used effectively for MPC [24].

## (iii) Identification of dynamics with feedback control

If the input $\mathbf{u}$ corresponds to feedback control, so that $\mathbf{u} = \mathbf{K}(\mathbf{x})$, then it is impossible to disambiguate the effect of the feedback control $\mathbf{u}$ with internal feedback terms $\mathbf{K}(\mathbf{x})$ within the dynamical system; namely, the SINDYc regression becomes ill-conditioned. In this case, we may identify the actuation $\mathbf{u}$ as a function of the state:

$$\mathbf{U} = \boldsymbol{\Xi}_u\boldsymbol{\Theta}^{\mathrm{T}}(\mathbf{X}). \tag{2.11}$$

To identify the coefficients $\boldsymbol{\Xi}$ in equation (2.4), we perturb the signal $\mathbf{u}$ to allow it to be distinguished from $\mathbf{K}(\mathbf{x})$ terms. This may be done by injecting a sufficiently large white noise signal, or occasionally kicking the system with a large impulse or step in $\mathbf{u}$. An interesting future direction would be to design input signals that *aid* in the identification of the dynamical system in equation (2.1) by perturbing the system in directions that yield high-value information.

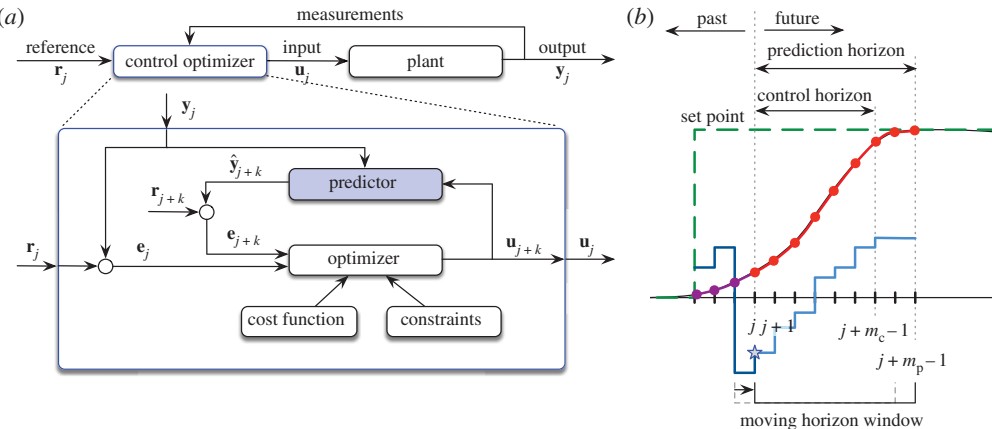

**Figure 4.** Schematics for (a) the control loop and (b) the receding horizon of MPC. Full state measurements $\mathbf{y} = \mathbf{x}$ and $\hat{\mathbf{y}} = \hat{\mathbf{x}}$ are considered as the output in the examples. Starting from the most recent measurement, the control input sequence (light blue solid) is optimized over the control horizon based on predicted future outputs (red solid) to drive the system to the set point (green dashed). The first input (blue star) in the sequence is enacted. (Online version in colour.)

## (b) Model predictive control

In this section, we outline the control problem and summarize key results in MPC, which is shown schematically in figure 4. MPC solves an optimal control problem over a receding horizon, subject to system constraints, to determine the next control action. This optimization is repeated at each new timestep, and the control law is updated, as shown in figure 4b.

The receding horizon control problem can generally be formulated as an open-loop optimization at each step, which determines the optimal sequence of control inputs $\mathbf{u}(\cdot|\mathbf{x}_j) := \{\mathbf{u}_{j+1}, \ldots, \mathbf{u}_{j+k}, \ldots, \mathbf{u}_{j+m_c}\}$ over the control horizon $T_c = m_c \Delta t$ given the current measurement $\mathbf{x}_j$ that minimizes a cost $J$ over the prediction horizon $T_p = m_p \Delta t$; $\Delta t$ is the timestep of the model, which may be different from the sampling time of measurements. The control horizon is generally less than or equal to the prediction horizon, so that $T_c \leq T_p$; if $T_c < T_p$, then the input $\mathbf{u}$ is assumed constant thereafter. The first control value $\mathbf{u}_{j+1}$ is then applied, and the optimization is reinitialized and repeated at each subsequent timestep to solve for the unknown sequence $\mathbf{u}(\cdot|\mathbf{x}_j)$. This results in an implicit feedback control law

$$\mathbf{K}(\mathbf{x}_j) = \mathbf{u}(j+1|\mathbf{x}_j) = \mathbf{u}_{j+1}, \tag{2.12}$$

where $\mathbf{u}_{j+1}$ is the first in the optimized actuation sequence starting at the initial condition $\mathbf{x}_j$.

The cost optimization at each timestep is given by

$$\min_{\hat{\mathbf{u}}(\cdot|\mathbf{x}_j)} J(\mathbf{x}_j) = \min_{\hat{\mathbf{u}}(\cdot|\mathbf{x}_j)} \left[ \|\hat{\mathbf{x}}_{j+m_p} - \mathbf{x}^*_{m_p}\|^2_{\mathbf{Q}_{m_p}} + \sum_{k=0}^{m_p-1} \|\hat{\mathbf{x}}_{j+k} - \mathbf{x}^*_k\|^2_{\mathbf{Q}} + \sum_{k=1}^{m_c-1} (\|\hat{\mathbf{u}}_{j+k}\|^2_{\mathbf{R}_u} + \|\Delta\hat{\mathbf{u}}_{j+k}\|^2_{\mathbf{R}_{\Delta u}}) \right], \tag{2.13}$$

subject to the discrete-time system dynamics with $\hat{\mathbf{F}} : \mathbb{R}^n \times \mathbb{R}^q \to \mathbb{R}^n$

$$\hat{\mathbf{x}}_{k+1} = \hat{\mathbf{F}}(\hat{\mathbf{x}}_k, \mathbf{u}_k), \tag{2.14}$$

the input constraints,

$$\Delta\mathbf{u}_{\min} \leq \Delta\mathbf{u}_k \leq \Delta\mathbf{u}_{\max} \tag{2.15a}$$

and

$$\mathbf{u}_{\min} \leq \mathbf{u}_k \leq \mathbf{u}_{\max}, \tag{2.15b}$$

and possibly additional equality or inequality constraints on the state and input. Here, we assume the availability of full-state measurements $\mathbf{y} = \mathbf{x}$. The cost functional $J$ penalizes deviations of the predicted state $\hat{\mathbf{x}}_k$ along the trajectory $\mathbf{x}_k^*$ and also includes a terminal cost at $\hat{\mathbf{x}}_{m_p}$. Expenditures of the input $\mathbf{u}_k$ and input rate $\Delta\mathbf{u}_k = \mathbf{u}_k - \mathbf{u}_{k-1}$ are also penalized. Each term is computed as the weighted norm of a vector, i.e. $\|\mathbf{x}\|_{\mathbf{Q}}^2 := \mathbf{x}^T\mathbf{Q}\mathbf{x}$. The weight matrices $\mathbf{Q} \geq 0$, $\mathbf{Q}_{m_p} \geq 0$, $\mathbf{R}_u > 0$ and $\mathbf{R}_{\Delta u} > 0$ are positive definite and positive semi-definite, respectively. Note that the model prediction $\hat{\mathbf{x}}_k$, which is forecast, may differ from the true measured state $\mathbf{x}_k$.

The dynamics are given by the identified SINDYc model, e.g. $\dot{\mathbf{x}} = \mathbf{F}(\mathbf{x}, \mathbf{u}) = \boldsymbol{\Xi}\boldsymbol{\Theta}^T(\mathbf{x}, \mathbf{u})$; $\hat{\mathbf{F}}$ represents a discrete-time or discretized SINDYc model. While the model and the control law may be learned simultaneously, we adopt a two-stage process, where the model is first learned from data and then used in the control optimization with MPC. A joint optimization of the model and the control law may be challenging, as the particular control action depends on the model. However, it may be possible to develop a streaming algorithm to adapt the model to abrupt system changes [30], iterating between model identification and control optimization.

MPC is one of the most powerful model-based control techniques due to the flexibility in the formulation of the objective functional, the ability to add constraints, and extensions to nonlinear systems. The most challenging aspect of MPC involves the identification of a dynamical model that accurately and efficiently represents the system behaviour when control is applied. If the model is linear, minimization of a quadratic cost functional subject to linear constraints results in a tractable convex problem. Nonlinear models may yield significant improvements; however, they render MPC a nonlinear program, which can be expensive to solve, making it particularly challenging for real-time control. Conditions on the well-posedness of the problem and existence and uniqueness of the solution of the nonlinear optimization problem are, e.g. provided in [60]. Fortunately, improvements in computing power and advanced algorithms are increasingly enabling nonlinear MPC for real-time applications.

## 3. A simple model for population dynamics

We first demonstrate the SINDY-MPC architecture on the Lotka–Volterra system, a two-dimensional, weakly nonlinear dynamical system, describing the interaction between two competing populations. These dynamics may represent two species in biological systems, competition in stock markets [61], and can be modified to study the spread of infectious diseases [62]. We will consider more sophisticated examples in the following sections.

The dynamics of the prey and predator populations, $x_1$ and $x_2$, respectively, are given by

$$\dot{x}_1 = ax_1 - bx_1x_2 \tag{3.1a}$$

and

$$\dot{x}_2 = -cx_2 + dx_1x_2 + u, \tag{3.1b}$$

where the constant parameters $a = 0.5$, $b = 0.025$, $c = 0.5$ and $d = 0.005$ represent the growth/death rates, the effect of predation on the prey population, and the growth of predators based on the size of the prey population. The unforced system exhibits a limit cycle behaviour, where the predator lags the prey, and a critical point $\mathbf{x}^{\text{crit}} = (g/d\ a/b)^T$, where the population sizes of both species are in balance. The control objective is to stabilize this fixed point. Here, the timestep $\Delta t = 0.1$ of the system and the model are equal, the weight matrices are $\mathbf{Q} = \left(\begin{smallmatrix} 1 & 0 \\ 0 & 1 \end{smallmatrix}\right)$ and $R_u = R_{\Delta u} = 0.5$, and the actuation input is limited to $u \in [-20, 20]$. The control and prediction horizons are $m_p = m_c = 10$ unless otherwise noted. We apply an additional constraint on $u$, so that $x_2$ does not decrease below 10, to enforce a minimum population size required for recovery.

To assess the performance and capabilities of the SINDY-MPC architecture, SINDYc is compared with two representative data-driven models: dynamic mode decomposition with control (DMDc) and a multilayer NN, which can represent any continuous function under mild conditions [63]. The results are displayed in figure 5. The first 100 time units are used to train the models with a phase-shifted sum of sinusoids as input, a so-called Schroeder sweep [64],

rspa.royalsocietypublishing.org  *Proc. R. Soc. A* **474**: 20180335

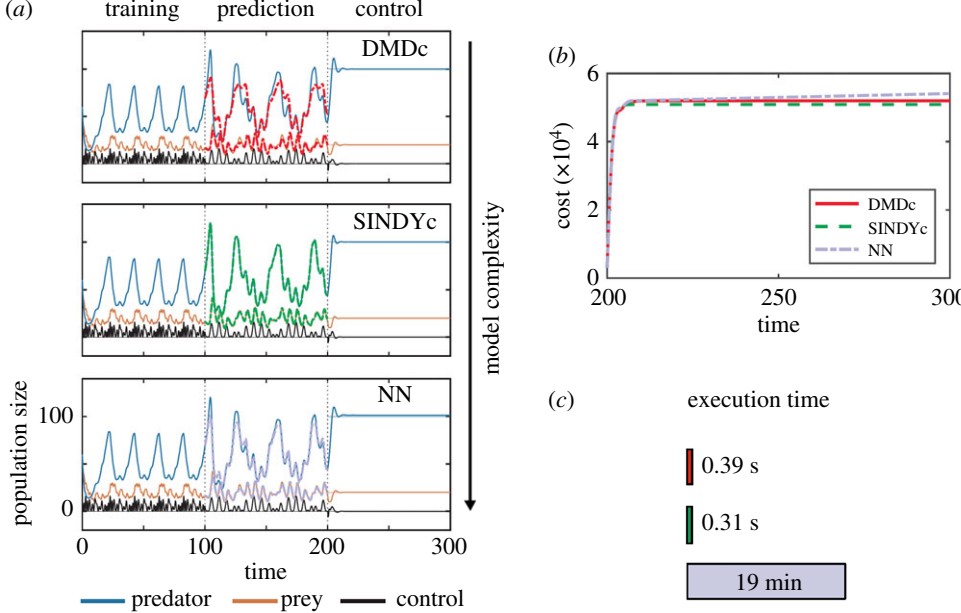

**Figure 5.** Prediction and control performance for the Lotka–Volterra system: (*a*) time series of states and input during training, validation and control stage, (*b*) cumulative cost and (*c*) execution time of the MPC optimization procedure. (Online version in colour.)

after which the predictive capabilities of these models are validated using sinusoidal forcing with $u(t) = (2\sin(t)\sin(t/10))^2$ on the next 100 time units. Different actuation inputs are used during the training and validation stages to assess the models' ability to generalize. Thereafter, MPC is applied for 100 time units using a prediction and control horizon of $m_{\mathrm{p}} = m_{\mathrm{c}} = 5$. SINDYc shows the best prediction and control performance, followed by DMDc and the NN (due to its steady-state error). The NN has 1 hidden layer with 10 neurons, which is the best trade-off between model complexity and accuracy; increasing the number of neurons or layers has little impact on the prediction performance. Further, hyperbolic tangent sigmoid activation functions are employed. It is first trained as a feed-forward network using the Levenberg–Marquardt algorithm and then closed. If the data are corrupted by noise, a Bayesian regularization is employed, which requires more training time but improves robustness. While the NN exhibits a similar control performance, the execution time of SINDYc is 37 times faster, which is particularly critical in real-time applications. For a fair comparison, all methods are compared using the same optimization routine based on interior-point methods via Matlab's `fmincon`. Thus, it would be possible to reduce the time for the linear system further.

In practice, measurements are generally affected by noise. We examine the robustness of these models for increasing noise corruption of the state measurements, i.e. $\mathbf{y} = \mathbf{x} + \mathbf{n}$ where $\mathbf{n} \in \mathcal{N}(0, \sigma^2)$ with standard deviation $\sigma$. Cross-validated prediction performance for different noise magnitudes $\eta = \sigma / \max(\mathrm{std}(x_i)) \in (0.01, 0.5)$, where std denotes standard deviation, is displayed in figure 7a,b. As expected, the performance of all models decreases with increasing noise magnitude. SINDYc generally outperforms DMDc and NN models, exhibiting a slower decline in performance for low and moderate noise levels. Sparse regression is known to improve robustness to noise and prevent overfitting. The large fluctuation in the NN performance are due to its strong dependency on the initial network weights.

The amount of data required to train an accurate model is particularly crucial in real-time applications, where abrupt changes or actuation may render the model invalid and rapid model updates are necessary. Figure 6a–c shows the average relative prediction error on 100 time units used for validation, and the training time for increasing lengths of training data. The effect of the

rspa.royalsocietypublishing.org *Proc. R. Soc. A* **474**: 20180335

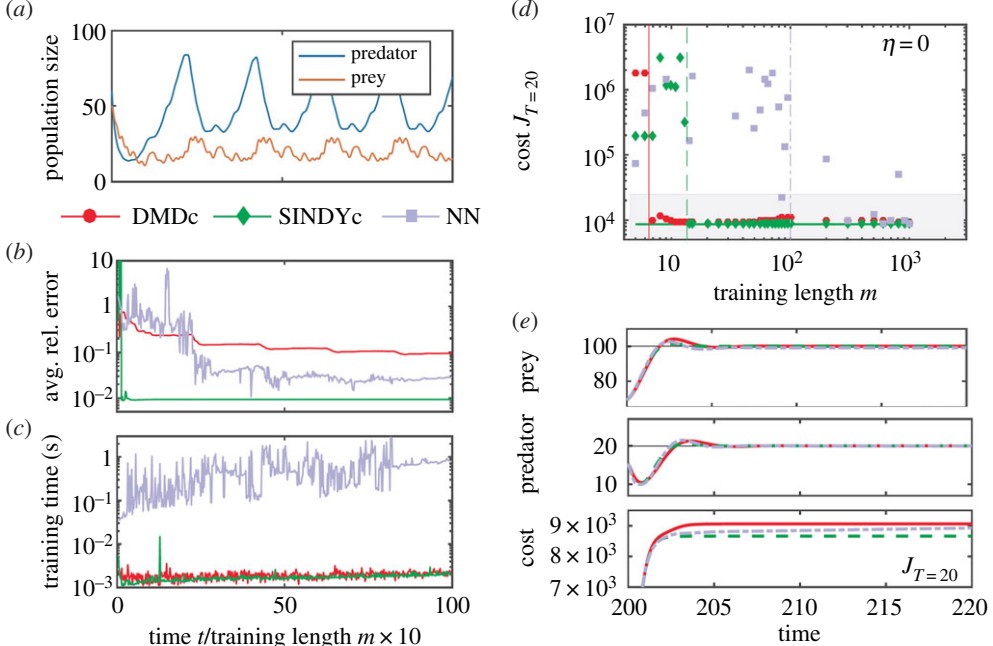

**Figure 6.** Cross-validated prediction error and control performance for increasing length of training data: (*a*) training time series, (*b*) average relative prediction error, (*c*) model training time in seconds, (*d*) terminal cumulative cost over 20 time units and (*e*) time series of states and cost of the best model for each model type ($m^{DMDc} = 20$, $m^{SINDYc} = 85$, $m^{NN} = 10^3$). From $m = 14$ onwards, SINDYc yields highly performing models in MPC, outperforming all other models. Outside the shaded region, models perform significantly worse or even diverge. (Online version in colour.)

training length on the control performance (evaluated over 20-time units) is shown in figure 6*d,e*. For small amounts of data, the sparsity-promoting parameter $\lambda$ in SINDYc is reduced by a factor of 10 until a non-zero entry appears. In the low-data limit, a highly predictive SINDYc model can be learned, discovering the true governing equations within machine precision. Significantly larger amounts of data are required to train an accurate NN model, although with enough data it outperforms DMDc. DMDc models may be useful in the extremely low-data limit, before enough data is available to characterize a SINDYc model. The training times of SINDYc and DMDc models increase slightly with the amount of data, but they require about two orders of magnitude less time than NN models. SINDYc's intrinsic robustness to overfitting renders all models from $m_{train} = 14$ on as having the best control performance compared with the overall best performing DMDc and NN models. By contrast, DMDc shows a slight decrease in performance due to overfitting and the NN's dependency on the initial network weights detrimentally affects its performance. It is interesting to note that the control performance is generally less sensitive than the long-term prediction performance shown in figure 6*b,c*. Even a model with moderately low predictive accuracy may perform well in MPC.

In figure 7*c,d*, we show the same analysis but with noise-corrupted training data. We assume no noise corruption during the control stage. For each training length, the best model out of 50 noise realizations is tested for control. DMDc and SINDYc models both require slightly more data to achieve a similar performance as without noise. Note that NN models perform significantly worse when trained on noise-corrupted data.

## 4. Chaotic Lorenz system

In this section, we demonstrate the SINDY-MPC architecture on the chaotic Lorenz system, a prototypical example of chaos in dynamical systems. The Lorenz system represents the

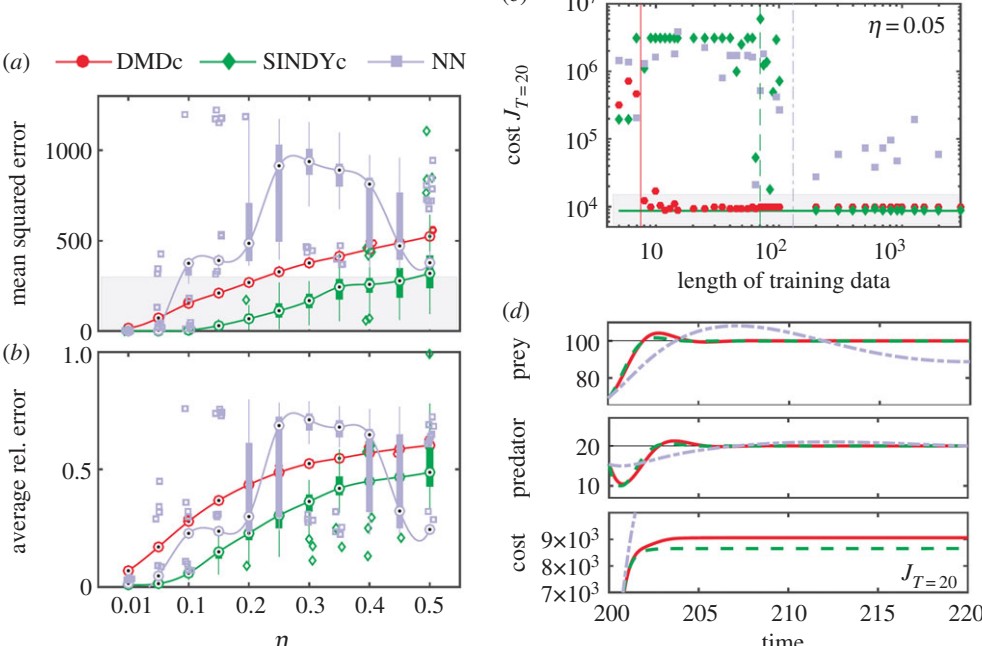

**Figure 7.** Cross-validated prediction and control performance for increasing measurement noise: (*a*) mean squared and (*b*) average relative prediction error, (*c*) terminal cumulative cost over 20 time units and (*d*) time series of states and cost of the best models ($m^{\text{DMDc}} = 12$, $m^{\text{SINDYc}} = 1250$, $m^{\text{NN}} = 65$). The control performance is shown for increasing length of noise-corrupted training data with $\eta = 0.5$. From $m = 200$ onwards, SINDYc yields highly performing models, outperforming all other models. Statistics are shown for 50 noise realizations each. Above the shaded region, models in most realizations do not have any predictive power. (Online version in colour.)

Rayleigh–Bénard convection in fluid dynamics as proposed by Lorenz [65], but has also been associated with lasers, dynamos and chemical reaction systems. The Lorenz dynamics are given by

$$\dot{x}_1 = \sigma(x_2 - x_1) + u \tag{4.1a}$$

$$\dot{x}_2 = x_1(\rho - x_3) - x_2 \tag{4.1b}$$

and

$$\dot{x}_3 = x_1 x_2 - \beta x_3 \tag{4.1c}$$

with system parameters $\sigma = 10$, $\beta = 8/3$, $\rho = 28$, and control input $u$ affecting only the first state. A typical trajectory oscillates alternately around the two weakly unstable fixed points $(\pm\sqrt{72}, \pm\sqrt{72}, 27)^{\text{T}}$. The chaotic motion of the system implies a strong sensitivity to initial conditions, i.e. small uncertainties in the state will grow exponentially with time. This represents a particularly challenging problem for model identification and subsequent control, as measurement and model uncertainty both lead to long-time forecast error.

The control objective is to stabilize one of these fixed points. In general, the timestep of the model is chosen to balance the control horizon, the length of the sequence of control inputs to be optimized and prediction accuracy. Here, the system timestep is $\Delta t^{\text{sys}} = 0.001$ and the model timestep is $\Delta t^{\text{model}} = 0.01$. The control input is determined every 10 system timesteps and then held constant. The weight matrices are $\mathbf{Q} = \mathbf{I}_3$, where $\mathbf{I}_n$ denotes a $n \times n$ identity matrix, $R_u = R_{\Delta u} = 0.001$, and the actuation input is limited to $u \in [-50, 50]$. The control and prediction horizon is $m_{\text{p}} = m_{\text{c}} = 10$ and the sparsity-promoting parameter in SINDYc is $\lambda = 0.1$, unless otherwise noted. For all cases, we assume access to full-state information.

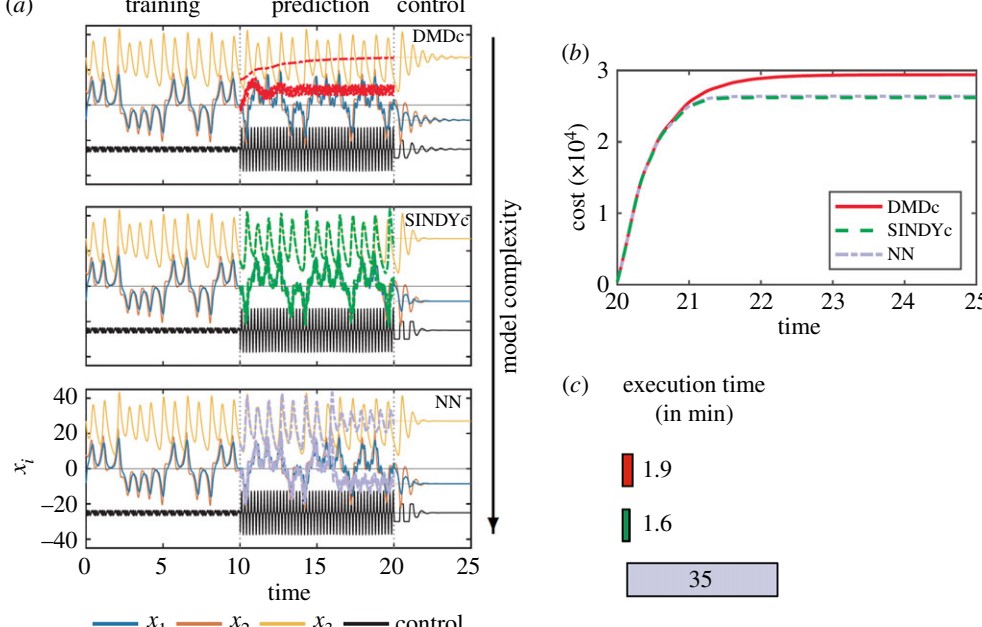

**Figure 8.** Prediction and control performance for the chaotic Lorenz system: (*a*) time series of the states and input (shifted to −25 and scaled by 10 to improve readability) during training, validation and control stage, (*b*) cumulative cost and (*c*) execution time of the MPC optimization. (Online version in colour.)

We compare the prediction and control performance of the SINDYc model with DMDc and NN models. DMDc is trained to model the deviation from the goal state by constructing the regression model based on data from which the goal state has been subtracted. A less naive approach would partition the trajectory into two bins, e.g. based on negative and positive values of $x_1$, and estimate two models for each goal state separately. The NN consists of 1 hidden layer with 10 neurons and employs hyperbolic tangent sigmoid activation functions. Cross-validated prediction and control performance for the Lorenz system are displayed in figure 8. The first 10 time units are used to train with a Schroeder sweep, after which the models are validated on the next 10 time units using a sinusoidally based high-frequency forcing, $u(t) = (5 \sin(30t))^3$. MPC is then applied for the last 5 time units. SINDYc exhibits the best prediction and control performance. The NN exhibits comparable control performance, although the prediction horizon is considerably shorter. Surprisingly, DMDc is able to stabilize the fixed point, despite poor predictions based on a linear model. As the predictive capability of DMDc is poor, we will not present DMDc results in the following, but instead compare SINDYc and the NN. As in the previous example, while the NN exhibits similar control performance, the control execution of SINDYc is 21 times faster.

Figure 9 examines the cross-validated prediction performance of SINDYc and NN models based on measurements with increasing noise magnitude $\eta = \sigma / \max(\mathrm{std}(x_i)) \in \{0.01, 0.1, 0.25\}$. The performance of both models decreases with increasing noise level, although SINDYc generally outperforms the NN. Unlike the Lotka–Volterra model, the average relative error is misleading in this case. With increasing noise magnitude the NN converges to a fixed point, having no predictive power, while SINDYc still exhibits the correct statistics beyond the prediction horizon; however, a phase drift leads to a larger average relative error. This is shown in figure 9b with the median (thick line) and the 25–75 percentile region (shaded area) of the prediction for three different noise levels. Thus, a better metric for prediction performance is the prediction horizon itself (figure 9a). The prediction horizon is estimated as the time instant when the error ball is larger than a radius of $\varepsilon = 3$, i.e. a model is considered predictive if

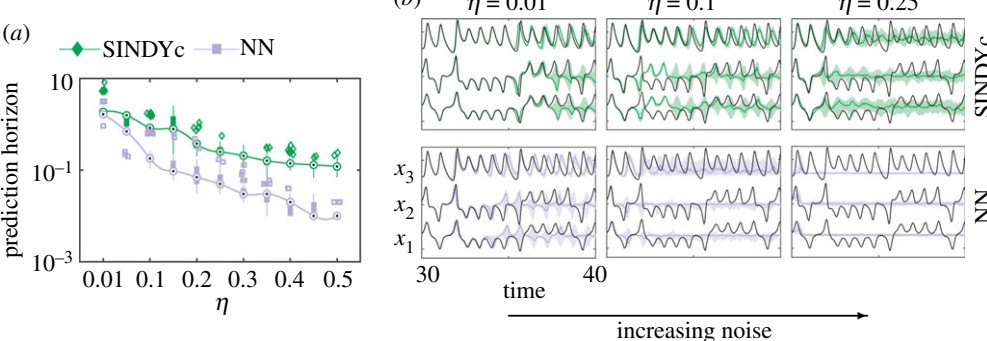

**Figure 9.** Cross-validated prediction performance for increasing measurement noise: (*a*) prediction horizon in time units and (*b*) time series with 50 (median as thick coloured line) and 25–75 (coloured shaded region) percentiles. Statistics are shown for 50 noise realizations each and different noise levels $\eta \in \{0.01, 0.1, 0.25\}$. (Online version in colour.)

rspa.royalsocietypublishing.org *Proc. R. Soc. A* **474**: 20180335

$\sqrt{\sum_{i=1}^{3}(x_i - \hat{x}_i)^2} < \varepsilon$. This corresponds to roughly 10% error per state variable, considering that the order of magnitude of each state is approximately $\mathcal{O}(10^1)$; this error radius correlates well with the visible divergence of the true and predicted state in figure 9*b*. For low and moderate noise levels, SINDYc robustly predicts the state with high accuracy. Even for $\eta = 0.25$, the 1-period prediction is sufficiently long for a successful stabilization with MPC as we consider a comparably short prediction horizon of $T_\mathrm{p} = 0.1$.

The effect of the amount of training data on the prediction and control performance is examined in figure 10, respectively. In figure 10*a–d*, we show the average relative error evaluated on the prediction over the next 10 time units, the prediction horizon, and the required training time in seconds for increasing length of noise-free training data. For a relatively small amount of data, SINDYc rapidly outperforms the NN model with a prediction horizon of 2.5 time units and a significantly smaller error. For a sufficiently large amount of data, SINDYc and the NN result in comparable predictions. However, SINDYc yields highly predictive models that can be rapidly trained in low and moderate data regimes. Models trained on weakly noise-corrupted measurements, $\eta = 0.05$, are tested in MPC. For each length of training data, 50 noise realizations are performed and the most predictive model is selected for evaluation in MPC (figure 10*e,f*). Outside the shaded regions, models are generally not predictive or might even diverge. In the noise-corrupted case, it is clear that SINDYc models generally have better control performance than NN models. For a sufficiently large amount of training data, NNs can have comparable performance to SINDYc models, although they show a sensitive dependence on the initial choice of the network weights. The control results of the NN are significantly better here than for the Lotka–Volterra model due to the intrinsic system properties. In chaotic systems, a long enough trajectory will come arbitrarily close to every point on the attractor; thus, measurements of the Lorenz system are in some sense richer than those of the Lotka–Volterra system. A surprising result is that a nearly optimal SINDYc model can be trained on just eight noisy measurements (compare figure 10*e,f*).

## 5. Tracking for the F-8 crusader

In this section, we consider an automatic flight control system of the F-8 aircraft at an altitude of 30 000 ft (9000 m) and Mach = 0.85 [66–68]. The control objective is to track a specific trajectory of

rspa.royalsocietypublishing.org *Proc. R. Soc. A* **474**: 20180335

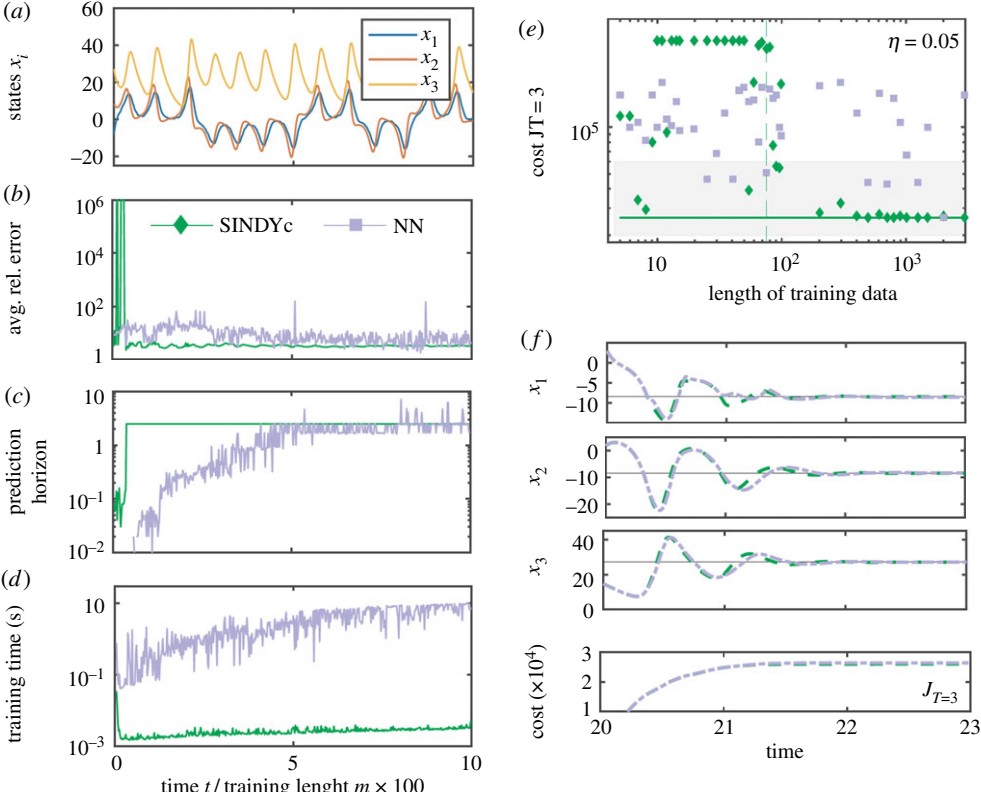

**Figure 10.** Cross-validated prediction and control performance for increasing length of training data (without noise): (*a*) time series of the training data, (*b*) average relative prediction error, (*c*) prediction horizon, (*d*) training time in seconds, (*e*) terminal cumulative cost over 3 time units and (*f*) time series of states and cost of the best model for each model type ($m^{\text{SINDYc}} = 38$, $m^{\text{NN}} = 40$). Note that from $m = 400$ onwards, SINDY identifies the best performing models. (Online version in colour.)

the angle of attack. The aircraft dynamics [66] are given by

$$\dot{x}_1 = -0.877x_1 + x_3 - 0.088x_1x_3 + 0.47x_1^2 - 0.019x_2^2 - x_1^2x_3 + 3.846x_1^3 - 0.215u$$
$$+ 0.28x_1^2u + 0.47x_1u^2 + 0.63u^3 \tag{5.1a}$$

$$\dot{x}_2 = x_3 \tag{5.1b}$$

and $\quad \dot{x}_3 = -4.208x_1 - 0.396x_3 - 0.47x_1^2 - 3.564x_1^3 - 20.967u + 6.265x_1^2u + 46x_1u^2 + 61.1u^3 \tag{5.1c}$

where $x_1$ is the angle of attack (rad), $x_2$ is the pitch angle (rad), $x_3$ is the pitch rate (rad s$^{-1}$) and $u$ is the control input representing the tail deflection angle (rad). The system is non-affine in the states and the control input rendering it strongly nonlinear. The commanded angle of attack to be tracked [68] is given by

$$r(t) = 0.4 \left( -\frac{0.5}{1 + e^{\hat{t}-0.8}} + \frac{1}{1 + e^{\hat{t}-3}} - 0.4 \right) \tag{5.2}$$

with $\hat{t} = t/0.1$. We assume that the output, over which the performance is optimized, is $y = x_1$. The timestep of the system is $\Delta t = 0.001$ and the timestep of the model is $\Delta t^M = 0.01$. The control input is determined using SINDY-MPC every 10 system timesteps over which the applied control is then kept constant. The weight matrices are $Q = 25$, $R_u = R_{\Delta u} = 0.05$, the actuation input rate is limited to $\Delta u \in [-0.3, 0.5]$, and the constraint for the angle of attack is $y \in [-0.2, 0.4]$. The control and prediction horizon is $m_p = m_c = 13$ and the sparsity-promoting parameter in SINDYc

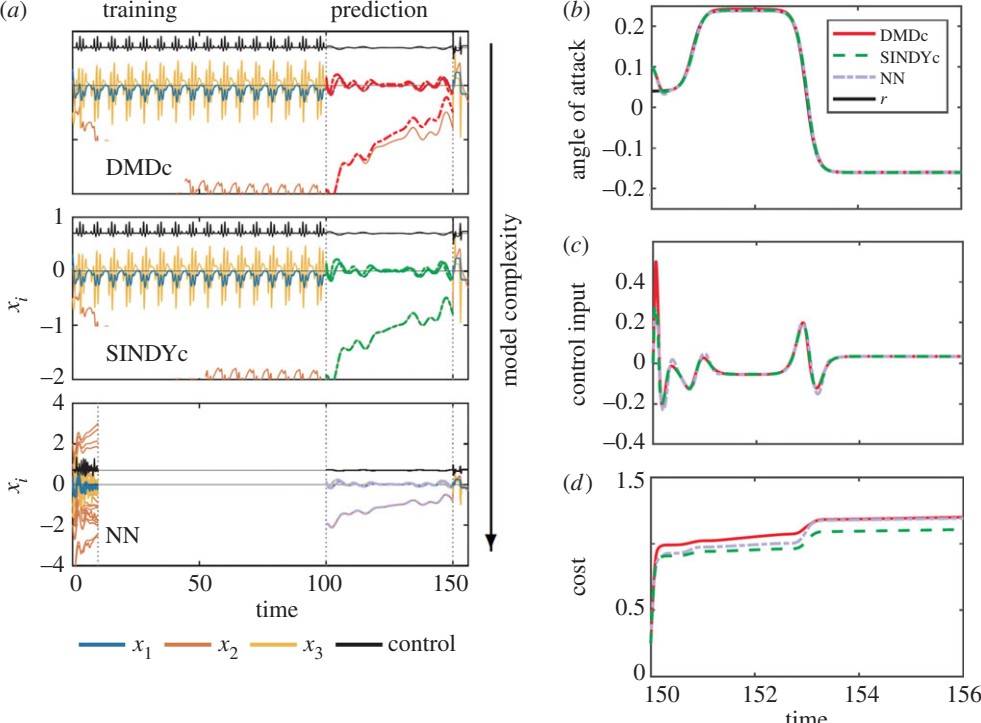

**Figure 11.** Prediction and control performance for F8 aircraft: (*a*) time series of states and input during training, validation and control, (*b*) angle of attack $y = x_1$ with reference *r*, (*c*) control input and (*d*) cumulative cost during control. (Online version in colour.)

is $\lambda = (10^{-4}, 10^{-2}, 10^{-2})$, where $\lambda_i$ is used to identify the terms for $x_i$. The NN has two hidden layers each with 15 neurons. Access to full-state information is assumed for these models.

Results assessing prediction and control performance of SINDYc compared with DMDc and a NN model are displayed in figure 11. Similar to the Lotka–Volterra system, the NN requires more and *richer* training data, i.e. a better exploration of the system behaviour, to perform sufficiently well. Thus, 250 short trajectories each consisting of 1000 snapshots ($25 \times 10^4$ instances in total) with varying input signals are used to train the NN; a subset of 20 trajectories is displayed in figure 11*a*(bottom). By contrast, SINDYc and DMDc perform similarly well if trained on much less data ($10^4$ instances of a single trajectory). Moreover, SINDYc learns from few measurements the true relationship between the variables, even though only limited system behaviour has been observed, resulting in increased performance.

# 6. Optimal therapy for pathogenic attacks

Optimizing drug therapy is critical for inhibiting diseases such as cancer and viral infections. Here, we consider treatment of infections with the human immunodeficiency virus (HIV), a pathogen that infects T-helper CD4+ cells of the immune system and can cause acquired immune deficiency syndrome (AIDS). Identifying the underlying infection mechanism, the response of the immune system, and the interactions with drugs targeting different components in this system is critical for developing and optimizing therapeutic strategies. Various models have been proposed to study the interaction between HIV and CD4+ cells; we refer to a recent review [69].

Optimal treatment aims to decrease virus mutations, complications from administered drugs, medical costs, and to strengthen the immune system. We consider a system [70] that incorporates

rspa.royalsocietypublishing.org Proc. R. Soc. A 474: 20180335

infections with HIV, the cytotoxic lymphocyte (CTL) response of the immune system, and therapeutic interventions via a highly active anti-retroviral therapy (HAART), i.e. a combination of drugs that affect the replication rate of HIV and support the immune system. This is based on a more general and complex system, that can be simplified under certain conditions [71]:

$$\dot{x}_1 = \lambda - dx_1 - \beta(1 - \eta u)x_1 x_2 \tag{6.1a}$$

$$\dot{x}_2 = \beta(1 - \eta u)x_1 x_2 - ax_2 - p_1 x_4 x_2 - p_2 x_5 x_2 \tag{6.1b}$$

$$\dot{x}_3 = c_2 x_1 x_2 x_3 - c_2 q x_2 x_3 - b_2 x_3 \tag{6.1c}$$

$$\dot{x}_4 = c_1 x_2 x_4 - b_1 x_4 \tag{6.1d}$$

and

$$\dot{x}_5 = c_2 q x_2 x_3 - h x_5 \tag{6.1e}$$

with parameters $\lambda = 1$, $d = 0.1$, $\beta = 1$, $a = 0.2$, $p_1 = 1$, $p_2 = 1$, $c_1 = 0.03$, $c_2 = 0.06$, $b_1 = 0.1$, $b_2 = 0.01$, $q = 0.5$, $h = 0.1$ and $\eta = 0.9799$ (units typically in $mm^{-3} d^{-1}$). Here, the states describe concentrations of healthy CD4+ T-cells, $x_1$, HIV-infected CD4+ T-cells, $x_2$, CTL precursors (memory CTL), $x_3$, helper-independent CTL, $x_4$ and helper-dependent CTL, $x_5$. For a detailed discussion of the system (6.1) we refer to [70,71]. The parameter $\eta$ represents the effectiveness of the HAART therapy applied via $u$. For the considered parameters and in the absence of control ($u \equiv 0$), the system exhibits two stable fixed points: a progressive infection leading to AIDS, $\mathbf{x}^A$, and the recovery from a successful immune response, $\mathbf{x}^B$. The later steady state is given by

$$x_1^B = \frac{\lambda}{d + \beta x_2^B}, \quad x_2^B = \frac{c_2(\lambda - dq) - b_2\beta - \sqrt{[c_2(\lambda - dq) - b_2\beta]^2 - 4\beta c_2 q d b_2}}{2\beta c_2 q} \tag{6.2a}$$

and

$$x_3^B = \frac{h x_5^B}{c_2 q x_2^B}, \quad x_4^B = 0, \quad x_5^B = \frac{x_2^B c_2(\beta q - a) + b_2\beta}{c_2 p_2 x_2^B}, \tag{6.2b}$$

and exists if $[c_2(\lambda - dq) - b_2\beta]^2 - 4\beta c_2 q d b_2 \geq 0$. The region of attraction (ROA) to this fixed point is limited and only established if the infectivity of the virus is small such that $\beta < c_1[c_2 b_2(\lambda - qd) - b_2 c_1 d]/b_1(c_2 b_1 q + b_2 c_1)$, which can be achieved by applying a HAART therapy ($u = 1$) with high efficacy ($\eta \approx 1$). This state moves as a function of $\beta_{\text{eff}} = \beta(1 - \eta u)$ when $u > 0$ and its ROA changes and does not necessarily overlap with the ROA in the absence of drug treatment ($u = 0$), i.e. dependent on the initial condition and the applied control the system will converge to a different steady state. A non-trivial control strategy is required that switches between treatment and no treatment to establish a successful immune response, and hence to approach $\mathbf{x}^B$. By contrast, when treatment is applied continuously for a sufficiently long amount of time such that the fixed points are approached and then terminated, the system will converge to a progressive infection, $\mathbf{x}^A$, even if a successful immune response had been established.

The cost functional to be optimized is given by

$$J = \int_0^T (x_1(t) - \hat{x}_1) + (x_3(t) - \hat{x}_3) + |u(t)| \, dt, \tag{6.3}$$

where $\hat{x}_1 = x_1^B$ and $\hat{x}_3 = x_3^B$ [70] taking into account the healthy cells, the immune system, and the cost of treatment. The control input $u$ is bounded by $0 \leq u \leq 1$ with efficacy of $\eta = 0.9799$. An additional constraint is added to the control that renders all cell concentrations non-negative, i.e. $x_i \geq 0 \, \forall i$. The time step is $\Delta t^M = 2 \, h$ for the model and is $\Delta t = 1/24 \, \text{day} = 1 \, h$ for the simulated system. The control performance is evaluated over 50 weeks. The prediction and control horizon for the MPC optimization are both $m_p = m_c = 24$, i.e. over 2 days (from $m_p \Delta t^M$). We assume a more realistic situation, where the state is measured once a week, and the treatment is then kept constant over the following week. The training data consists of samples collected over 200 days ($\approx 30$ weeks) with a discrete control input, as was applied for the validation data in figure 12 (bottom). By contrast, the training data for the NN consists of ensemble data of 32 different

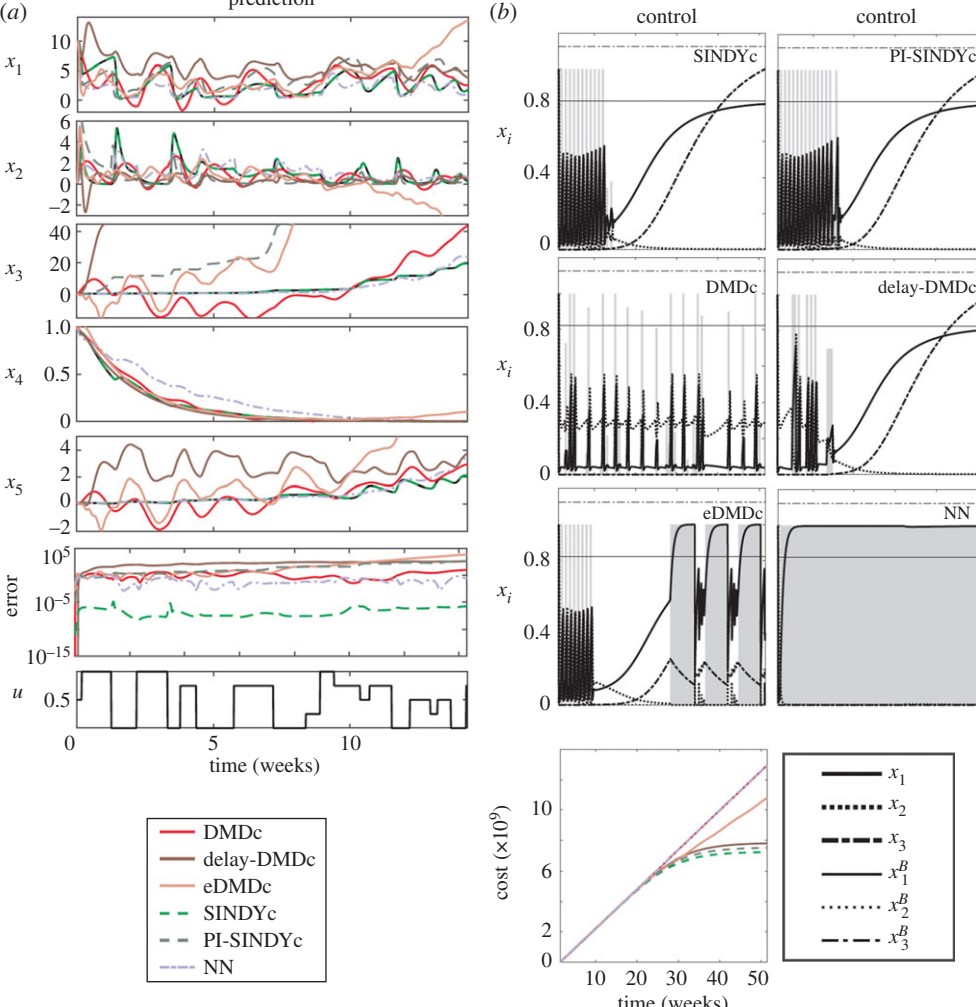

**Figure 12.** Prediction and control performance for various models of the HIV model: (*a*) Predicted and true (solid black) time series of the forced dynamics and (*b*) control results (normalized for better legibility) with optimized actuation input (shaded background) and reference values. Only SINDYc and, with slightly less performance, PI-SINDYc and delay-DMDc are capable of driving the system to the desired steady state. (Online version in colour.)

trajectories. In both cases, the control is a random sequence of values that are kept constant over random durations of time $\Delta T \in [5\,\mathrm{h}, 10\,\mathrm{days}]$.

We consider a SINDYc model with (1) full-state information (SINDYc) and (2) partial information based on a subset of the variables (PI-SINDYc). The latter case demonstrates the situation when only a few states can be measured, which is generally more realistic. For the identification of the SINDYc models, it is important to normalize first the features in the library, as the coefficients of the active terms spread over several orders of magnitude. In both cases, a polynomial order of three is used for the library. The results are compared with various linear models: (1) DMDc on the full state (DMDc), (2) DMDc on delay coordinates of the full state (Delay-DMDc) and (3) DMDc on a set of nonlinear observables (extended DMD with control, eDMDc). In addition, a NN model on the full state (NN) is trained. An overview of these models and their parameters is provided in table 1. The identified parameters of the SINDYc and PI-SINDYc

rspa.royalsocietypublishing.org  Proc. R. Soc. A 474: 20180335

**Table 1.** Model parameters for the HIV system.

| model | state | settings | # unknowns |
|---|---|---|---|
| SINDYc | $\mathbf{y} = \mathbf{x}$ | polynomial basis (order $r = 3$), $\boldsymbol{\lambda} = (10, 3.1, 3, 0.1, 0.5)$ | $84 \times 5 = 420$ |
| PI-SINDYc | $\mathbf{y} = (x_1, x_2, x_3)$ | polynomial basis (order $r = 3$), $\boldsymbol{\lambda} = (10, 30, 3)$ | $35 \times 3 = 105$ |
| DMDc | $\mathbf{y} = \mathbf{x} - \mathbf{x}^B$ | deviation from reference state | $5^2 = 25$ |
| Delay-DMDc | $\mathbf{y} = (\mathbf{x}(t) - \mathbf{x}^B, \mathbf{x}(t - \tau) - \mathbf{x}^B, \ldots, \mathbf{x}(t - 9\tau) - \mathbf{x}^B)$ | deviation from reference state, 10 time delay coordinates of the full-state and of the control input | $(10 \times 5)^2 = 2500$ |
| eDMDc | $\mathbf{y} = \boldsymbol{\Theta}(\mathbf{x}, r)$ | order $r = 3$ of polynomial basis (without constant term) | $83^2 = 6889$ |
| NN | $\mathbf{y} = \mathbf{x}$ | 1 hidden layer with 5 neurons and linear activation functions, data is *log*-transformed and mapped to $[-1, 1]$ to compensate for skewness and different range | 43 |

models are displayed below (6.4) are:

$$
\underbrace{\begin{bmatrix}
0.9995 & 0 & 0 & 0 & 0 \\
-0.0999 & 0 & 0 & 0 & 0 \\
0 & -0.1991 & 0 & 0 & 0 \\
0 & 0 & -0.0100 & 0 & 0 \\
0 & 0 & 0 & -0.1000 & 0 \\
0 & 0 & 0 & 0 & -0.1000 \\
-0.9990 & 0.9981 & 0 & 0 & 0 \\
0 & 0 & -0.0299 & 0 & 0.0300 \\
0 & -0.9982 & 0 & 0.0300 & 0 \\
0 & -0.9990 & 0 & 0 & 0 \\
0 & 0 & 0.0600 & 0 & 0 \\
0 & 0 & 0 & 0 & 0 \\
0 & 0 & 0 & 0 & 0 \\
0.9763 & -0.9757 & 0 & 0 & 0
\end{bmatrix}}_{\boldsymbol{\Xi}^{SINDYc}}
\quad
\underbrace{\begin{bmatrix}
0.9995 & 0 & 0 \\
-0.0999 & 0 & 0 \\
0 & -0.8985 & 0 \\
0 & 0 & -0.0100 \\
0 & 0 & 0 \\
0 & 0 & 0 \\
-0.9990 & 0.9424 & 0 \\
0 & -0.0573 & -0.0299 \\
0 & 0 & 0 \\
0 & 0 & 0 \\
0 & 0.0069 & 0.0600 \\
0 & 0 & 0 \\
0 & 0 & 0 \\
0.9763 & -0.7507 & 0
\end{bmatrix}}_{\boldsymbol{\Xi}^{PI-SINDYc}}
\begin{matrix}
1 \\
x_1 \\
x_2 \\
x_3 \\
x_4 \\
x_5 \\
x_1 x_2 \\
x_2 x_3 \\
x_2 x_4 \\
x_2 x_5 \\
x_1 x_2 x_3 \\
x_1 x_2 x_4 \\
x_1 x_2 x_5 \\
x_1 x_2 u
\end{matrix}
$$

$$(6.4)$$

Only the non-zero parameters are shown, and their error is $\mathcal{O}(10^{-3}) - \mathcal{O}(10^{-6})$ for SINDYc. The error in the parameters decreases with increased time resolution. Here, a coarse time step is chosen to reduce the computational cost of MPC for the chosen prediction horizon. In PI-SINDYc, the parameters for $x_1$ and $x_3$ are estimated well as these only depend on $x_1$, $x_2$ and $x_3$. By contrast, $x_2$ has a larger error in the estimated parameters and consists of erroneous parameters to compensate for the missing information. Different selections of variables have been tested, which generally resulted in poor models, except for the selected combination. The resulting models are generally not sparse, except where a direct relationship exists between variables. This suggests that SINDY indicates direct causal relationships, which can be measured in terms of the sparsity.

Prediction accuracy based on data differing from the training set, but with a similar type of actuation signal, and control results are displayed in figure 12. Both start from an early infection given by $\mathbf{x}_0 = (\lambda/d, 0.1, 0.1, 0.1, 0.1)^{\mathrm{T}}$. While a SINDYc model can be identified with near-perfect prediction accuracy, all other models display an error several orders of magnitude larger (see figure 12a). In particular, linear DMDc-based models diverge significantly from the true trajectory for some variables, while capturing the right trend in other variables. The NN and the PI-SINDYc model based on partial state information generally stay closer to, and even temporarily match, the true trajectory. Interestingly, while MPC using PI-SINDYc successfully drives the system to the desired steady-state behaviour, with a slightly larger cost than SINDYc, the NN controller is unable to establish the successful immune response by applying constant treatment (figure 12b). Note that the actuation depends strongly on the prediction and control horizon chosen for the

rspa.royalsocietypublishing.org Proc. R. Soc. A 474: 20180335

**Table 2.** Capabilities and challenges of DMDc, SINDYc and NN models. The model with the strongest performance is underlined.

| property | DMDc | SINDYc | NN |
|---|---|---|---|
| training with limited data | **strong** | **strong** | **weak** |
| | very few samples are sufficient | well suited for low and medium amount of data | requires long time series to learn predictive models |
| high-dimensionality | **strong** | **fair** | **strong** |
| | can handle high-dim. data in combination with SVD | limited by the library size | |
| nonlinearities | **weak**/fair | **strong** | **strong** |
| | linear and weakly nonlinear, however with performance loss | suitable for strongly nonlinear systems | suitable for strongly nonlinear systems |
| prediction performance | **fair** | **strong** | **strong** |
| Control performance | **fair** | **strong** | **strong** |
| noise robustness | **weak** | **strong** | **fair** |
| | high sensitivity w.r.t. noise | intrinsic robustness due to sparse regression | can handle low noise levels |
| parameter robustness | **strong** | **strong** | **weak** |
| | | | high sensitivity w.r.t. initial weights of the network |
| training time | **strong** | **strong** | **weak** |
| execution time | **strong** | **strong** | **weak** |
| | fast optimization routines exist for linear systems | | |

optimization; further analysis has shown that a smaller horizon for the NN controller yields a time-varying, however still unsuccessful, treatment. We varied the number of hidden layers (up to 3), the number of neurons (up to 100), the type of activation function, the number of delays (up to 100) in the state and input variables and the amount of training data ($\approx 600$ different initial conditions). However, these did not significantly change the performance of the model. The type of data (not just the amount) is particularly critical for training a NN. Designing experiments, i.e. a good forcing signal that explores the system behaviour and yields *dynamically rich* training data, is a challenge of its own. The linear DMDc and eDMDc models fail too. While the eDMDc model starts with the correct frequency, detrimental treatment is administered thereafter close to the desired state, which gives rise to new growth of infected cells, $x_2$. Interestingly, augmenting the state vector with delay coordinates results in a successful treatment (with performance close to the SINDYc models), in contrast to the strategy to augment the state with nonlinear measurements of the state as in eDMDc.

All models but the NN, which has been trained on a significantly larger amount of data, have been trained on the same amount of data, a single trajectory starting from an initial condition which is relatively far from the desired behaviour. Thus, these models are required to generalize well, i.e. perform well far from the region in which they have been initially trained. Using more data would certainly help to improve the prediction accuracy of some of these models, in particular, if these require a large number of parameters to be estimated. However, this would pose additional challenges in real-time applications with abrupt system changes, as this requires robust model formation and adaptation from few measurements.

# 7. Discussion and conclusion

In conclusion, we have demonstrated the effective integration of data-driven sparse model discovery for MPC in the low-data limit. The sparse identification of nonlinear dynamics (SINDY) algorithm has been extended to discover nonlinear models with actuation and control, resulting in interpretable and parsimonious models. Moreover, because SINDY only identifies the few active terms in the dynamics, it requires less data than many other leading machine learning techniques, such as NNs, and prevents overfitting. When integrated with MPC, SINDY provides computationally tractable and accurate models that can be trained on very little data. The resulting SINDY-MPC framework is capable of controlling strongly nonlinear systems, purely from measurement data, and the model identification is fast enough to discover models in real-time, even in response to abrupt changes to the model. The SINDY-MPC approach is compared with MPC based on data-driven linear models and NN models on four nonlinear dynamical systems of different complexities and challenges: the weakly nonlinear Lotka–Volterra system, the chaotic Lorenz system, the non-affine F8 crusador model, and the HIV/immune response system, which variables are of different order of magnitudes and where only partial state information is available.

The relative strengths and weaknesses of each method are summarized in table 2. By nearly every metric, linear DMDc models and nonlinear SINDYc models outperform NN models. In fact, DMDc may be seen as the limit of SINDYc when the library of candidate terms is restricted to linear terms. SINDY-MPC provides the highest performance control and requires significantly less training data and execution time compared with NN. However, for very low amounts of training data, DMDc provides a useful model until the SINDYc algorithm has enough data to characterize the dynamics. Thus, we advocate the SINDY-MPC framework for effective and efficient nonlinear control, with DMDc as a stopgap after abrupt changes until a new SINDYc model can be identified. Note that a crucial step in SINDY is the choice of library functions, which is often informed by expert knowledge about what category of nonlinearities to include. A poor choice of the library will generally yield a non-sparse model. Without any prior knowledge about the system type, a sweep through different classes of candidate functions is required. However, once a model is learned from a sufficiently rich library, the model is often able to generalize beyond the training data. If the model structure is not fixed, but varies heterogeneously in state space, NNs may provide a more flexible and generalizable architecture to represent the dynamics. A heterogeneous model structure can potentially be incorporated into SINDy by additionally learning a library of models [72,73].

This work motivates a number of future extensions and investigations. Although the preliminary application of SINDYc for MPC is encouraging, this study does not leverage many of the powerful new techniques in sparse model identification. Figure 3 provides a schematic of the modularity and demonstrated extensions that are possible within the SINDy framework. In realistic applications, the system may be extremely high-dimensional, and the SINDy library does not scale well with the size of the data. Fortunately, many high-dimensional systems evolve on a low-dimensional attractor, and it is often possible to identify a model on this attractor, for example by identifying a SINDy model on low-dimensional coordinates obtained through a singular value decomposition [2] or manifold learning [74]. In other applications, full-state measurements are unavailable, and the system must be characterized by limited measurements. It has recently been shown that delay coordinates provide a useful embedding to identify simple models of chaotic systems [53], building on the celebrated Takens embedding theorem [75]. Delay coordinates also define intrinsic coordinates for the Koopman operator [53], which provides a simple linear embedding of nonlinear systems [76,77]. Koopman models have recently been used for MPC [24,25] and have been identified using SINDy regression [78] and subsequently used for optimal control [78]. Recently, SINDY has been extended to modify an existing model based on new incoming measurements to enable rapid model recovery from abrupt changes to the system [30]. Learning quickly from limited measurements is an important task, which may be viewed in terms of design of experiments; specifically, optimizing the actuation input to collect

rspa.royalsocietypublishing.org *Proc. R. Soc. A* **474**: 20180335

the most informative measurements to learn a more predictive model faster. This would require the formulation of a different cost function, which measures the predictive power of the model, to determine future actuation inputs. Rapid learning is also related to the question of quantity versus quality of data and identifiability [48,49]; more data is usually better, although it is possible to work with less data if it is representative of the system. Further, similar methods could be used to optimize sensors and exploit partial measurements within the SINDY-MPC framework. All of these innovations suggest a shift from the perspective of *big data* to the control-oriented perspective of *smart data*.

Figure 3 also demonstrates innovations to the SINDy regression to include physical constraints, known model structure, and model selection, which may all benefit the goal of real-time identification and control. Known symmetries, conservation laws, and constraints may be readily included in both the SINDYc and DMDc modelling frameworks [31], as they are both based on least-squares regression, possibly with sequential thresholding. It is thus possible to use a constrained least-squares algorithm, for example, to enforce energy conserving constraints in a fluid system, which manifest as anti-symmetric quadratic terms [31]. Enforcing constraints has the potential to further reduce the amount of data required to identify models, as there are less free parameters to estimate, and the resulting systems have been shown to have improved stability in some cases. It is also possible to extend the SINDy algorithm to identify models in libraries that encode richer dynamics, such as rational function nonlinearities [79]. Finally, incorporating information criteria provides an objective metric for model selection among various candidate SINDy models with a range of complexity.

The SINDY-MPC framework has significant potential for the real-time control of strongly nonlinear systems. Moreover, the rapid training and execution times indicate that SINDy models may be useful for rapid model identification in response to abrupt model changes, and this warrants further investigation. The ability to identify accurate and efficient models with small amounts of training data may be a key enabler of recovery in time-critical scenarios, such as model changes that lead to instability. In addition, for broad applicability and adoption, the SINDy modelling framework must be further investigated to characterize the effect of noise, derive error estimates, and provide conditions and guarantees of convergence. These future theoretical and analytical extensions are necessary to certify the model-based control performance.

Data accessibility. The code used in this work is made available at: https://github.com/eurika-kaiser/SINDY-MPC. The data can be generated using the code.

Authors' contributions. All authors conceived of the work, designed the study and drafted the manuscript. E.K. carried out the computations.

Competing interests. We declare we have no competing interests.

Funding. E.K. gratefully acknowledges support by the Washington Research Foundation, the Gordon and Betty Moore Foundation (Award no. 2013-10-29), the Alfred P. Sloan Foundation (Award no. 3835), and the University of Washington eScience Institute. S.L.B. and J.N.K. acknowledge support from the Defense Advanced Research Projects Agency (DARPA contract HR011-16-C-0016 and PA-18-01-FP-125). S.L.B. acknowledges support from the Army Research Office (W911NF-17-1-0306 and W911NF-17-1-0422). J.N.K. acknowledges support from the Air Force Office of Scientific Research (FA9550-17-1-0329).

Acknowledgements. The authors gratefully acknowledge many valuable discussions with Josh Proctor.

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
