## [Reviewer comments · Proceedings. Mathematical, Physical, and Engineering Sciences]

Review History

RSPA-2018-0335.R0 (Original submission)

Review form: Referee 1 (Russel Caflisch)

Is the manuscript an original and important contribution to its field?

Yes

Is the paper of sufficient general interest?

Yes

Is the overall quality of the paper suitable?

Yes

Quality of the paper

An excellent paper making an important contribution to the field: should be published.

Can the paper be shortened without overall detriment to the main message?

No

Do you think some of the material would be more appropriate as an electronic appendix?

No

For papers with colour figures – is colour essential?

Yes

If there is supplementary material, is this adequate and clear?

Not applicable

Are there details of how to obtain materials and data, including any restrictions that may apply?

No

Do you have any ethical concerns with this paper?

No

Recommendation?

Accept with minor revision (please list in comments)

Comments to the Author(s)

This is a very interesting manuscript on a timely and important problem, the performance of different methods for learning, prediction and control of nonlinear systems. The manuscript compares sparse identification of nonlinear dynamics (SINDY), which was developed by the authors, with two other methods - dynamic mode decomposition (DMD) and neural nets (NN) - on four interesting model problems. On these models, SINDY performs significantly better than the other two methods, especially in terms of the amount of data required to train the method, the computational time required, and the ability to predict behavior outside the range of training data.

This work is commendable for its careful description of the applications and the performance of these competing methods on those applications. This is illuminating for anyone who is trying to apply machine learning and similar techniques in new applications. For these reasons, this manuscript is a useful addition to the field.

My only doubt about the study described here is whether it is a fair comparison. The model applications to which the methods are applied consist of ODEs $dx/dt = f(x)$, with right-hand sides $f(x)$ that are polynomial in the variables. This is an ideal problem for SINDY, since SINDY relies on use of a library of functions, with polynomials being the most natural candidates. Moreover, the statement that SINDY can make predictions outside the range of training data, is somewhat misleading. Although the behavior of the system may be different in different regions of phase space $\{x\}$, the form of the model is uniform in space, since $f(x)$ involves a small number of polynomial terms. So once it has been learned in one region of phase space, it will be valid everywhere.

NNs have no bias toward polynomials, and they have been most effective on problems that have a multiscale or heterogeneous structure. If the function $f(x)$ had a different polynomial form in different regions of phase space, then the NNs might very well outperform SINDY.

This does not invalidate the results of the study, but the bias and limitations of the study should be clearly and prominently acknowledged.

I have opted for open peer reviewing, because I'm curious about it, not because I advocate for it.

Russ Caflisch, Courant Institute, New York University

Review form: Referee 2

Is the manuscript an original and important contribution to its field?

Yes

Is the paper of sufficient general interest?

Yes

Is the overall quality of the paper suitable?

Yes

Quality of the paper

A paper that may be acceptable after major revision.

Can the paper be shortened without overall detriment to the main message?

Yes

Do you think some of the material would be more appropriate as an electronic appendix?

No

For papers with colour figures – is colour essential?

No

If there is supplementary material, is this adequate and clear?

Yes

Are there details of how to obtain materials and data, including any restrictions that may apply?

Yes

Do you have any ethical concerns with this paper?

No

Recommendation?

Major revision is needed (please make suggestions in comments)

Comments to the Author(s)

The authors incorporate a recent sparse identification of nonlinear dynamics modeling procedure in a model predictive control framework (SINDY-MPC). The proposed framework has been investigated on various dynamics whose governing equations are multivariate polynomials. It's shown from the examples that the new model requires less data empirically, and is more computationally efficient and robust to noise than 1-layer neural network. The authors have an interesting point of view about the predictive control problems. The examples are investigated thoroughly. Could the authors verify the following questions?

1. The authors may want to check the relevant between the citations and the related comments about those works. For example,
 - a. In the introduction, lines 32-34 page 3, "These challenges point to the need ...that may be

characterized from limited data and in response to abrupt changes". It's not clear why the contents of those papers have connections with limited data and abrupt changes.

b. In Section 2, lines 29-32 page 4, "... compressed sensing has been used to handle noise and outliers [34]..." It seems that the work [34] discussed about linear systems which doesn't fit into the subsection "Sparse identification of nonlinear dynamics with control".

2. About the tensor notation in Equation 2.3: Based on the definition of $X \otimes X$ and $X \otimes U$, it seems that $X \otimes X$ will include both $x^1 x^2$ and $x^2 x^1$, where $x = (x^1, x^2, \dots, x^n)$. As a consequence, the matrix Θ^T will have repeated rows. Does it affect the optimization problem (2.4) in terms of well-posedness and/or overfitting?

3. Equation (2.4) is inconsistent with the equation at the top right corner of Figure 3. The dimensions of the left and right sides of Equation (2.4) don't match.

4. There is something wrong with Equation (2.5). Could the authors explain how to solve Equation (2.5) using LASSO or the cited sequentially thresholded least square procedure? Please also check the consequence formulas.

5. Is the optimization problem (2.5) well-posed? In the literature, there are various conditions to guarantee the uniqueness of the LASSO problem. For the discussed problems and examples in the paper, could the authors provide some rigorous analysis about conditions can be applied or extended from those of the sparse regression model, especially when the matrix Θ is built from both x and u while the right hand side consists only x ?

6. About the MPC model (Equation 2.13), what is the relation between u_k and \hat{u}_k ? What are the unknowns? What is \hat{F} ? Where is the place that the Equation (2.5) involves in the process (The authors may want to explain in words about Figure 2). How do the authors choose the parameter λ in Equation (2.5) along the SINDY-MPC process? Could the authors discuss about the well-posedness of the SINDY-MPC? Does it have a unique solution? How does it depend on the parameters?

7. In the comparisons in Sections 3,4, and 5, how the neural network solve the MPC problem, especially dealing with the constraints?

8. Based on the set up and examples, it seems to me that the authors consider the problem of finding a function, say G , of variable $Z = (x, u)$, subject to the constraints for x and u with weighted least square for Z and the gradient of u . What are the benefits of constructing under the model predictive control framework?

9. Finally, it seems that the title is a little bit misunderstanding. The low-data property is observed via numerical examples, not from the SINDY-MPC framework itself. The authors may want to explain more about the connection between low-data limit and the proposed framework either from modeling perspective or from theoretical point of view.

Decision letter (RSPA-2018-0335.R0)

24-Aug-2018

Dear Dr Kaiser

The Editor of Proceedings A has now received comments from referees on the above paper and would like you to revise it in accordance with their suggestions which can be found below (not including confidential reports to the Editor).

Please submit a copy of your revised paper within four weeks - if we do not hear from you within this time then it will be assumed that the paper has been withdrawn. In exceptional circumstances, extensions may be possible if agreed with the Editorial Office in advance.

Please note that it is the editorial policy of Proceedings A to offer authors one round of revision in

which to address changes requested by referees. If the revisions are not considered satisfactory by the Editor, then the paper will be rejected, and not considered further for publication by the journal. In the event that the author chooses not to address a referee's comments, and no scientific justification is included in their cover letter for this omission, it is at the discretion of the Editor whether to continue considering the manuscript.

- Ethics statement
- Data accessibility
- Competing interests
- Authors' contributions
- Acknowledgements
- Funding statement

See <http://royalsocietypublishing.org/instructions-authors#question3> for further details.

To revise your manuscript, log into <http://mc.manuscriptcentral.com/prsa> and enter your Author Centre, where you will find your manuscript title listed under "Manuscripts with Decisions." Under "Actions," click on "Create a Revision." Your manuscript number has been appended to denote a revision.

You will be unable to make your revisions on the originally submitted version of the manuscript. Instead, revise your manuscript and upload a new version through your Author Centre.

When submitting your revised manuscript, you will be able to respond to the comments made by the referee(s) and upload a file "Response to Referees" in "Section 6 - File Upload". Please use this to document how you have responded to the comments, and the adjustments you have made. In order to expedite the processing of the revised manuscript, please be as specific as possible in your response to the referee(s).

IMPORTANT: Your original files are available to you when you upload your revised manuscript. Please delete any unnecessary previous files before uploading your revised version.

When revising your paper please ensure that it remains under 28 pages long. In addition, any pages over 20 will be subject to a charge (£150 + VAT (where applicable) per page). Your paper has been ESTIMATED to be 24 pages.

Once again, thank you for submitting your manuscript to Proc. R. Soc. A and I look forward to receiving your revision. If you have any questions at all, please do not hesitate to get in touch.

Yours sincerely

Alice Power
Publishing Editor
Proceedings A
proceedingsa@royalsociety.org

Reviewer(s)' Comments to Author:

Referee: 1

Comments to the Author(s)

This is a very interesting manuscript on a timely and important problem, the performance of different methods for learning, prediction and control of nonlinear systems. The manuscript compares sparse identification of nonlinear dynamics (SINDY), which was developed by the authors, with two others methods - dynamic mode decomposition (DMD) and neural nets (NN) - on four interesting model problems. On these models, SINDY performs significantly better than the other two methods, especially in terms of the amount of data required to train the method, the computational time required, and the ability to predict behavior outside the range of training data.

This work is commendable for its careful description of the applications and the performance of these competing methods on those applications. This is illuminating for anyone who is trying to apply machine learning and similar techniques in new applications. For these reasons, this manuscript is a useful addition to the field.

My only doubt about the study described here is whether it is a fair comparison. The model applications to which the methods are applied consist of ODEs $dx/dt = f(x)$, with right-hand sides $f(x)$ that are polynomial in the variables. This is an ideal problem for SINDY, since SINDY relies on use of a library of functions, with polynomials being the most natural candidates. Moreover, the statement that SINDY can make predictions outside the range of training data, is somewhat misleading. Although the behavior of the system may be different in different regions of phase space $\{x\}$, the form of the model is uniform in space, since $f(x)$ involves a small number of polynomial terms. So once it has been learned in one region of phase space, it will be valid everywhere.

NNs have no bias toward polynomials, and they have been most effective on problems that have a multiscale or heterogeneous structure. If the function $f(x)$ had a different polynomial form in different regions of phase space, then the NNs might very well outperform SINDY.

This does not invalidate the results of the study, but the bias and limitations of the study should be clearly and prominently acknowledged.

I have opted for open peer reviewing, because I'm curious about it, not because I advocate for it.

Russ Caflisch, Courant Institute, New York University

Referee: 2

Comments to the Author(s)

The authors incorporate a recent sparse identification of nonlinear dynamics modeling procedure in a model predictive control framework (SINDY-MPC). The proposed framework has been investigated on various dynamics whose governing equations are multivariate polynomials. It's shown from the examples that the new model requires less data empirically, and is more computationally efficient and robust to noise than 1-layer neural network. The authors have an interesting point of view about the predictive control problems. The examples are investigated thoroughly. Could the authors verify the following questions?

1. The authors may want to check the relevant between the citations and the related comments about those works. For example,

a. In the introduction, lines 32-34 page 3, “These challenges point to the need ...that may be characterized from limited data and in response to abrupt changes”. It’s not clear why the contents of those papers have connections with limited data and abrupt changes.

b. In Section 2, lines 29-32 page 4, “... compressed sensing has been used to handle noise and outliers [34]...” It seems that the work [34] discussed about linear systems which doesn’t fit into the subsection “Sparse identification of nonlinear dynamics with control”.

2. About the tensor notation in Equation 2.3: Based on the definition of $X \otimes X$ and $X \otimes U$, it seems that $X \otimes X$ will include both $x^1 x^2$ and $x^2 x^1$, where $x = (x^1, x^2, \dots, x^n)$. As a consequence, the matrix Θ^T will have repeated rows. Does it affect the optimization problem (2.4) in terms of well-posedness and/or overfitting?

3. Equation (2.4) is inconsistent with the equation at the top right corner of Figure 3. The dimensions of the left and right sides of Equation (2.4) don’t match.

4. There is something wrong with Equation (2.5). Could the authors explain how to solve Equation (2.5) using LASSO or the cited sequentially thresholded least square procedure? Please also check the consequence formulas.

5. Is the optimization problem (2.5) well-posed? In the literature, there are various conditions to guarantee the uniqueness of the LASSO problem. For the discussed problems and examples in the paper, could the authors provide some rigorous analysis about conditions can be applied or extended from those of the sparse regression model, especially when the matrix Θ is built from both x and u while the right hand side consists only x ?

6. About the MPC model (Equation 2.13), what is the relation between u_k and \hat{u}_k ? What are the unknowns? What is \hat{F} ? Where is the place that the Equation (2.5) involves in the process (The authors may want to explain in words about Figure 2). How do the authors choose the parameter λ in Equation (2.5) along the SINDY-MPC process? Could the authors discuss about the well-posedness of the SINDY-MPC? Does it have a unique solution? How does it depend on the parameters?

7. In the comparisons in Sections 3,4, and 5, how the neural network solve the MPC problem, especially dealing with the constraints?

8. Based on the set up and examples, it seems to me that the authors consider the problem of finding a function, say G , of variable $Z = (x, u)$, subject to the constraints for x and u with weighted least square for Z and the gradient of u . What are the benefits of constructing under the model predictive control framework?

9. Finally, it seems that the title is a little bit misunderstanding. The low-data property is observed via numerical examples, not from the SINDY-MPC framework itself. The authors may want to explain more about the connection between low-data limit and the proposed framework either from modeling perspective or from theoretical point of view.

Author's Response to Decision Letter for (RSPA-2018-0335.R0)

See Appendices A & B.

RSPA-2018-0335.R1 (Revision)

Review form: Referee 1 (Russel Caflisch)

Is the manuscript an original and important contribution to its field?

Yes

Is the paper of sufficient general interest?

Yes

Is the overall quality of the paper suitable?

Yes

Quality of the paper

An excellent paper making an important contribution to the field: should be published.

Can the paper be shortened without overall detriment to the main message?

No

Do you think some of the material would be more appropriate as an electronic appendix?

No

For papers with colour figures - is colour essential?

Yes

If there is supplementary material, is this adequate and clear?

Yes

Are there details of how to obtain materials and data, including any restrictions that may apply?

Yes

Do you have any ethical concerns with this paper?

No

Recommendation?

Accept as is

Comments to the Author(s)

This revised article is ready for publication now.

Review form: Referee 2

Is the manuscript an original and important contribution to its field?

Yes

Is the paper of sufficient general interest?

Yes

Is the overall quality of the paper suitable?

Yes

Quality of the paper

An excellent paper making an important contribution to the field: should be published.

Can the paper be shortened without overall detriment to the main message?

No

Do you think some of the material would be more appropriate as an electronic appendix?

No

For papers with colour figures – is colour essential?

Yes

If there is supplementary material, is this adequate and clear?

Yes

Are there details of how to obtain materials and data, including any restrictions that may apply?

Yes

Do you have any ethical concerns with this paper?

No

Recommendation?

Accept as is

Comments to the Author(s)

I am happy with the changes the authors made. The paper should be published now.

Decision letter (RSPA-2018-0335.R1)

Dear Dr Kaiser

On behalf of the Editor, I am pleased to inform you that your manuscript entitled "Sparse identification of nonlinear dynamics for model predictive control in the low-data limit" has been accepted in its final form for publication in Proceedings A.

Our Production Office will be in contact with you in due course. You can expect to receive a proof of your article soon. Please contact the office to let us know if you are likely to be away from e-mail in the near future. If you do not notify us and comments are not received within 5 days of sending the proof, we may publish the paper as it stands.

Open access

You are invited to opt for open access, our author pays publishing model. Payment of open access fees will enable your article to be made freely available via the Royal Society website as soon as it is ready for publication. For more information about open access please visit http://royalsocietypublishing.org/site/authors/open_access.xhtml. The open access fee for this journal is £1700/\$2380/€2040 per article. VAT will be charged where applicable.

Note that if you have opted for open access then payment will be required before the article is published – payment instructions will follow shortly. If you wish to opt for open access then please inform the editorial office (proceedingsa@royalsociety.org) as soon as possible.

Your article has been estimated as being 24 pages long. Our Production Office will inform you of the exact length at the proof stage.

Proceedings A levies charges for articles which exceed 20 printed pages. (based upon approximately 540 words or 2 figures per page). Articles exceeding this limit will incur page charges of £150 per page or part page, plus VAT (where applicable).

Under the terms of our licence to publish you may post the author generated postprint (ie. your accepted version not the final typeset version) of your manuscript at any time and this can be made freely available. Postprints can be deposited on a personal or institutional website, or a recognised server/repository. Please note however, that the reporting of postprints is subject to a media embargo, and that the status the manuscript should be made clear. Upon publication of the definitive version on the publisher's site, full details and a link should be added.

You can cite the article in advance of publication using its DOI. The DOI will take the form: 10.1098/rspa.XXXX.YYYY, where XXXX and YYYY are the last 8 digits of your manuscript number (eg. if your manuscript number is RSPA-2017-1234 the DOI would be 10.1098/rspa.2017.1234).

For tips on promoting your accepted paper see our blog post:
<https://blogs.royalsociety.org/publishing/promoting-your-latest-paper-and-tracking-your-results/>

Thank you for your submission. On behalf of the Editors of the journal, we look forward to your continued contributions to the Journal.

Best wishes

Alice Power
Publishing Editor
Proceedings A Editorial Office
proceedingsa@royalsociety.org

Reviewer(s)' Comments to Author:

Referee: 1

Comments to the Author(s)
This revised article is ready for publication now.

Referee: 2

Comments to the Author(s)
I am happy with the changes the authors made. The paper should be published now.

Appendix A

Reply to Referee #2

Roy. Soc. Proc. A manuscript: **RSPA-2018-0335**

entitled "Sparse identification of nonlinear dynamics for model predictive control in the low-data limit" by EURIKA KAISER, J. NATHAN KUTZ & STEVEN L. BRUNTON

General comments and main changes

We highly appreciate the referees for their careful reviews and thoughtful suggestions. In considering their comments, we believe that we have improved the technical presentation and clarity of the manuscript. We have addressed each of the reviewer comments throughout the manuscript. Changes suggested by Referee #1 and #2 are highlighted in **magenta** and **blue**, respectively.

Specific comments to Referee #2

The authors incorporate a recent sparse identification of nonlinear dynamics modeling procedure in a model predictive control framework (SINDY-MPC). The proposed framework has been investigated on various dynamics whose governing equations are multivariate polynomials. It's shown from the examples that the new model requires less data empirically, and is more computationally efficient and robust to noise than 1-layer neural network. The authors have an interesting point of view about the predictive control problems. The examples are investigated thoroughly. Could the authors verify the following questions?

We are grateful for the referee for their thoughtful suggestions and remarks and their positive assessment. We have addressed the referee's specific questions below.

1. *The authors may want to check the relevant between the citations and the related comments about those works. For example,*
 - a. *In the introduction, lines 32-34 page 3, These challenges point to the need that may be characterized from limited data and in response to abrupt changes. Its not clear why the contents of those papers have connections with limited data and abrupt changes.*
 - b. *In Section 2, lines 29-32 page 4, compressed sensing has been used to handle noise and outliers [34] It seems that the work [34] discussed about linear systems which doesnt fit into the subsection Sparse identification of nonlinear dynamics with control.*

We thank the referee for pointing out possible reference mismatches. The papers in a) refer to interpretable models, but not to the second part of the sentence on limited data and abrupt changes. We have added a reference that extends SINDy to cope with abrupt

changes, which we believe makes this clearer. We think reference [34] mentioned in b) is important to keep as it discusses compressed sensing, a field that is very much related and its concepts are used for developing sparsity-promoting methods. While we agree that it doesn't discuss nonlinear models, it is very related from the sparse identification perspective.

2. *About the tensor notation in Equation 2.3: Based on the definition of $X \otimes X$ and $X \otimes U$, it seems that $X \otimes X$ will include both $x^1 x^2$ and $x^2 x^1$, where $x = (x^1, x^2, \dots, x^n)$. As a consequence, the matrix Θ^T will have repeated rows. Does it affect the optimization problem (2.4) in terms of well-posedness and/or overfitting?*

We thank the referee for pointing this out. This issue arises from trying to find a compact mathematical representation for the library. In practice we build the library using only unique terms, which can be efficiently implemented (without the need to construct all possible combinations). I suspect that identical columns would lead to an equal split of the corresponding coefficient; more generally, highly correlated columns lead to an ill-conditioned library. As the library size increases exponentially with the number of variables, it is not advisable to incorporate more terms than strictly necessary. We have added a comment to clarify that all redundant terms are omitted.

3. *Equation (2.4) is inconsistent with the equation at the top right corner of Figure 3. The dimensions of the left and right sides of Equation (2.4) don't match.*

Thanks for pointing this out; we fixed this in the figure.

4. *There is something wrong with Equation (2.5). Could the authors explain how to solve Equation (2.5) using LASSO or the cited sequentially thresholded least square procedure? Please also check the consequence formulas.*

We fixed the missing squaring in Eqs. (2.5) and (2.8) in the l_2 norm term. We also added pseudocode for the sequentially thresholding algorithm (Algorithm 1) that hopefully helps. Here, entries in $\hat{\Xi}$ are thresholded with respect to the parameter ε , for which Eq. (2.5) represents the Lagrangian form.

5. *Is the optimization problem (2.5) well-posed? In the literature, there are various conditions to guarantee the uniqueness of the LASSO problem. For the discussed problems and examples in the paper, could the authors provide some rigorous analysis about conditions can be applied or extended from those of the sparse regression model, especially when the matrix Θ is built from both x and u while the right hand side consists only x ?*

This is a very important question and we thank the referee for pointing this out. We have added a paragraph on recovery and convergence, which has been investigated in several

works. In particular, the conditions for convergence and recovery of a sparse vector in the l_1 relaxation problem have been provided in *Tropp, 2006 IEEE Trans. Inform. Theory* and *Su et al., arXiv 2016* [48,49]. The convergence of the sequentially thresholding least-squares algorithm has been analyzed in *Zhang & Schaeffer, arXiv 2018*[50] and the SINDy architecture appears also as a special case in the Sparse Relaxed Regularized Regression framework in *Zheng et al., arXiv 2018* [51]. In the context of system identification for control, i.e. where the right hand side depends on the state and input, we refer to recent work on identifiability of a polynomial nonlinear model structure (*Gevers, M. et al., 52nd IEEE Conference Decision and Control, 2013*) [78].

6. *About the MPC model (Equation 2.13), what is the relation between u_k and \hat{u}_k ? What are the unknowns?*

The u_k corresponds to one element of the optimal sequence that is returned by solving problem (2.13), which is optimized over \hat{u}_k , i.e. $u_k = \min_{\hat{u}_k} J(\mathbf{x}_j, \hat{\mathbf{u}}_k)$ is the found optimal solution (for simplicity we write here u_k , but it is actually a sequence of values denoted by $\mathbf{u}(\cdot|\mathbf{x}_j)$; these are the unknowns for which the optimization problem is solved). We fixed the notation in (2.13) and slightly changed the text to make this clearer.

What is \hat{F} ? Where is the place that the Equation (2.5) involves in the process (The authors may want to explain in words about Figure 2). How do the authors choose the parameter lambda in Equation (2.5) along the SINDY-MPC process?

\hat{F} represents the learned model through the SINDYc architecture, that prescribes the temporal evolution of the state. We added a clarification in the paragraph below Eq. (2.15). In general, the approach is a two-stage process: First, a model is learned using SINDYc. Then, the control optimization problem of MPC is solved, in which the model is used for prediction of the future state. The SINDY-MPC architecture is here not formulated as a joint optimization process over the model \hat{F} and the control sequence $\mathbf{u}(\cdot|\mathbf{x}_j)$. However, it is possible in this scheme to continuously update the SINDYc model \hat{F} , so that the scheme switches between learning and updating the model and optimizing the control input, e.g. in response to abrupt changes in the system. If the SINDYc model is learned offline from previously collected data, then a λ is chosen according to a pareto-optimal solution balancing model accuracy and complexity, e.g. see *Mangan et al., Proc. Roy. Soc. A 2017* [44]. Depending on the amount of data to analyze, this might not be feasible in a streaming version. But we also emphasize that a model update at each timestep/sampling time would not be necessary in most situations. In contrast, it is often sufficient (and computationally responsible) to update the model based on an auxiliary function which measures how much the model diverges from the sampled data, similarly as developed in *Quade et al., arXiv 2018* [30]. While the sparsity-promoting parameter λ is set here in advance, it would certainly be feasible to additionally optimize for λ , as the time between required updates of the model is much longer. We have added clarifications in the revised manuscript. Note also the next to last paragraph in Sec. 2(a) on the choice of λ .

Could the authors discuss about the well-posedness of the SINDY-MPC? Does it have a unique solution? How does it depend on the parameters?

We clarified in the revised manuscript that we do not solve a joint optimization problem for the model and the control, but follow a two-step process of (1) first learning the model in the SINDYc architecture and then (2) the control inputs solving the receding horizon problem. As also discussed briefly above, we added a paragraph on convergence and well-posedness of the SINDYc problem. In addition, there is a rich literature exploring this in the context of MPC and providing sufficient conditions. We have added relevant references in Sec. 2.

7. *In the comparisons in Sections 3,4, and 5, how the neural network solve the MPC problem, especially dealing with the constraints?*

We do not solve a joint optimization problem, but first learn the neural network model and then incorporate this in the MPC as a predictor for the future state. Thus, the neural network has not to deal necessarily with the constraints directly, except when providing the training data. For instance, the neural network will perform much worse in predicting accurately the future state based on input, if the input values lie outside of the range of values the neural network has been trained on.

8. *Based on the set up and examples, it seems to me that the authors consider the problem of finding a function, say G , of variable $Z = (x,u)$, subject to the constraints for x and u with weighted least square for Z and the gradient of u . What are the benefits of constructing under the model predictive control framework?*

This is aligned with the questions above addressing the joint optimization of model and controller. We are not doing this in the present work, but the optimization is separately performed for model discovery and control inputs, respectively. While a joint optimization may be constructed, it is not clear that this would be advantageous, as the control input optimization relies on a predictive model. However, an interesting direction would be to formulate this as a design of experiment to learn a good model. Specifically, optimizing the actuation sequence to collect informative measurements to learn (possibly faster) a (more predictive) model. This would require the formulation of a different cost function which measures the predictive power of the model. We have added clarifying comments in the manuscript.

9. *Finally, it seems that the title is a little bit misunderstanding. The low-data property is observed via numerical examples, not from the SINDY-MPC framework itself. The authors may want to explain more about the connection between low-data limit and the proposed framework either from modeling perspective or from theoretical point of view.*

This is very important question and we thank the referee for pointing this out. Indeed the title is not meant literally as taking the limit, although this would be an interesting

investigation. There are several aspects to consider when approaching the low-data limit: (1) there will be a minimum amount of data required (which will vary with different systems, probably dependent on the system's complexity), (2) in general, more data is required if measurements are corrupted with noise, (3) this also relates to questions of identifiability, e.g. what inputs $\mathbf{u}(t)$ need to be applied to collect the most informative data (which relates to requiring less data for the training process), (4) and the type of data, e.g. single trajectory versus ensemble data. Summarizing, this addresses the question of quantity and quality of data. More data is usually better; however, we can get away with less data if the data is generalizable to the system. A systematic analysis of this topic is very interesting and would require a notion of how informative data is in the system identification context. In a follow-up work we plan to investigate design-of-experiment questions, which relate to this topic; specifically, designing an actuation series to sample optimally informative data for a fast, i.e. from few samples, model discovery. We have added a brief discussion in Sec. 7 of the revised manuscript.

Appendix B

Reply to Referee #1

Roy. Soc. Proc. A manuscript: **RSPA-2018-0335**

entitled "*Sparse identification of nonlinear dynamics for model predictive control in the low-data limit*" by EURIKA KAISER, J. NATHAN KUTZ & STEVEN L. BRUNTON

General comments and main changes

We highly appreciate the referees for their careful reviews and thoughtful suggestions. In considering their comments, we believe that we have improved the technical presentation and clarity of the manuscript. We have addressed each of the reviewer comments throughout the manuscript. Changes suggested by Referee #1 and #2 are highlighted in **magenta** and **blue**, respectively.

Specific comments to Referee #1

This is a very interesting manuscript on a timely and important problem, the performance of different methods for learning, prediction and control of nonlinear systems. The manuscript compares sparse identification of nonlinear dynamics (SINDY), which was developed by the authors, with two others methods - dynamic mode decomposition (DMD) and neural nets (NN) - on four interesting model problems. On these models, SINDY performs significantly better than the other two methods, especially in terms of the amount of data required to train the method, the computational time required, and the ability to predict behavior outside the range of training data.

This work is commendable for its careful description of the applications and the performance of these competing methods on those applications. This is illuminating for anyone who is trying to apply machine learning and similar techniques in new applications. For these reasons, this manuscript is a useful addition to the field.

We are grateful for the referee's positive assessment of our work and appreciate the valid concern they raise, which is addressed below.

My only doubt about the study described here is whether it is a fair comparison. The model applications to which the methods are applied consist of ODEs $dx/dt = f(x)$, with right-hand sides $f(x)$ that are polynomial in the variables. This is an ideal problem for SINDY, since SINDY relies on use of a library of functions, with polynomials being the most natural candidates. Moreover, the statement that SINDY can make predictions outside the range of training data, is somewhat misleading. Although the behavior of the system may be different in different regions of phase space $\{x\}$, the form of the model is uniform in space,

since $f(x)$ involves a small number of polynomial terms. So once it has been learned in one region of phase space, it will be valid everywhere. NNs have no bias toward polynomials, and they have been most effective on problems that have a multiscale or heterogeneous structure. If the function $f(x)$ had a different polynomial form in different regions of phase space, then the NNs might very well outperform SINDY.

This does not invalidate the results of the study, but the bias and limitations of the study should be clearly and prominently acknowledged.

This is a very valid point, and we have modified the manuscript to more carefully caveat when SINDY is expected to work well and what assumptions are made throughout. Indeed, SINDY relies on a suitable choice of library functions, which also limits its application to some extent. The construction of the library is where expert knowledge enters the framework, which has benefits but also carries potential risks: If we know something about the system, e.g. the type of nonlinearities or type of relevant functions, we can easily incorporate these. We might also be biased and make false assumptions; however, this will most likely yield a non-sparse model, which is a clear indicator that the choice of candidate functions has been poor. If we don't know anything about the system, we would need, e.g., to sweep through classes of candidate functions, which can be cumbersome, limited, and result in scalability issues. We are actively working on overcoming scalability problems by employing tensor decomposition techniques. There exist also works on combining the SINDY architecture with neural networks for feature engineering, on which sparse dynamics are then learned.

However, all these developments will face issues with the mentioned problem, where the state space is partitioned into fundamentally different dynamics or an exogenous input drives these system changes. Neural networks may indeed generalize better in the heterogeneous model situation, and we add this important point to the discussion. In this case it may also be possible to combine SINDY with a library of dynamics for these different regimes (e.g. in state space or parameter-driven) as proposed in *Brunton et al. SIADS, 2014*. Another way to address this issue is the rapid re-learning of the SINDYc model as new measurements become available from other state space regions, although this may fail to capture sharp transitions in the dynamics. In general, we do not wish to discourage the use of neural networks. They have been proven to be incredibly successful in many applications including control, particularly for high-dimensional problems. However, they generally do not provide new insights into the physics of the problem, which could then be exploited, and require a (sometimes prohibitively) large amount of training data. In contrast, the SINDY(c) architecture yields interpretable models that can be analyzed further, from relatively few measurements, which is particularly crucial if the system "suffers" from abrupt changes & varying conditions. But there is certainly not a one-model-fits-all solution and we appreciate the referee pointing out this aspect which leads to the more subtle points when comparing different modeling strategies. We have added additional comments on this topic in Sec. 7 of the revised manuscript.